# β-cell deletion of the PKm1 and PKm2 isoforms of pyruvate kinase in mice reveals their essential role as nutrient sensors for the K$_{ATP}$ channel

Hannah R Foster[1], Thuong Ho[1], Evgeniy Potapenko[1], Sophia M Sdao[1], Shih Ming Huang[1], Sophie L Lewandowski[1], Halena R VanDeusen[1], Shawn M Davidson[2,3], Rebecca L Cardone[4], Marc Prentki[5], Richard G Kibbey[4,6], Matthew J Merrins[1,7]*

[1]Department of Medicine, Division of Endocrinology, Diabetes, and Metabolism, University of Wisconsin-Madison, Madison, United States; [2]Koch Institute for Integrative Cancer Research, Massachusetts Institute of Technology, Cambridge, United States; [3]Lewis-Sigler Institute for Integrative Genomics, Princeton University, Princeton, United States; [4]Department of Internal Medicine, Yale University, New Haven, United States; [5]Molecular Nutrition Unit and Montreal Diabetes Research Center, CRCHUM, and Departments of Nutrition, Biochemistry and Molecular Medicine, Université de Montréal, Montréal, Canada; [6]Department of Cellular & Molecular Physiology, Yale University, New Haven, United States; [7]William S. Middleton Memorial Veterans Hospital, Madison, United States

*For correspondence:
merrins@wisc.edu

Competing interest: The authors declare that no competing interests exist.

**Abstract** Pyruvate kinase (PK) and the phosphoenolpyruvate (PEP) cycle play key roles in nutrient-stimulated K$_{ATP}$ channel closure and insulin secretion. To identify the PK isoforms involved, we generated mice lacking β-cell PKm1, PKm2, and mitochondrial PEP carboxykinase (PCK2) that generates mitochondrial PEP. Glucose metabolism was found to generate both glycolytic and mitochondrially derived PEP, which triggers K$_{ATP}$ closure through local PKm1 and PKm2 signaling at the plasma membrane. Amino acids, which generate mitochondrial PEP without producing glycolytic fructose 1,6-bisphosphate to allosterically activate PKm2, signal through PKm1 to raise ATP/ADP, close K$_{ATP}$ channels, and stimulate insulin secretion. Raising cytosolic ATP/ADP with amino acids is insufficient to close K$_{ATP}$ channels in the absence of PK activity or PCK2, indicating that K$_{ATP}$ channels are primarily regulated by PEP that provides ATP via plasma membrane-associated PK, rather than mitochondrially derived ATP. Following membrane depolarization, the PEP cycle is involved in an 'off-switch' that facilitates K$_{ATP}$ channel reopening and Ca$^{2+}$ extrusion, as shown by PK activation experiments and β-cell PCK2 deletion, which prolongs Ca$^{2+}$ oscillations and increases insulin secretion. In conclusion, the differential response of PKm1 and PKm2 to the glycolytic and mitochondrial sources of PEP influences the β-cell nutrient response, and controls the oscillatory cycle regulating insulin secretion.

## Editor's evaluation

This manuscript employs in vitro studies and elegant mouse models to detail how specific pyruvate kinase isoforms impact pancreatic β-cell ATP/ADP levels, ATP-sensitive K$^+$ channel (KATP channel) activity, calcium handling, and insulin secretion. This is an important study that challenges the current paradigms of KATP-channel regulation, the major signaling mechanism through which

pancreatic β cells couple blood glucose levels to insulin release. Future studies will determine whether similar mechanisms are used in human pancreatic β cells.

## Introduction

Maintenance of euglycemia relies on β-cells to couple nutrient sensing with appropriate insulin secretion. Insulin release is stimulated by the metabolism-dependent closure of ATP-sensitive K$^+$ (K$_{ATP}$) channels (*Ashcroft et al., 1984*; *Cook and Hales, 1984*; *Misler et al., 1986*; *Rorsman and Trube, 1985*), which triggers Ca$^{2+}$ influx and exocytosis (*Anderson and Long, 1947*; *Grodsky et al., 1963*). Contrary to what is often believed, the glucose-induced signaling process in β-cells has not been largely solved, and the entrenched model implicating a rise in mitochondrially derived ATP driving K$_{ATP}$ channel closure (*Campbell and Newgard, 2021*; *Prentki et al., 2013*) is incomplete and possibly wrong, the main reason being that it does not consider other sources of local ATP production that may be key for signaling (*Corkey, 2020*; *Lewandowski et al., 2020*; *Merrins et al., 2022*). The recent discovery that pyruvate kinase (PK), which converts ADP and phosphoenolpyruvate (PEP) to ATP and pyruvate, is present on the β-cell plasma membrane where it is sufficient to raise sub-plasma membrane ATP/ADP (ATP/ADP$_{pm}$) and close K$_{ATP}$ channels (*Lewandowski et al., 2020*) provides an alternative mechanism to oxidative phosphorylation for K$_{ATP}$ channel regulation. Based on this finding, *Lewandowski et al., 2020*, proposed a revised model of β-cell fuel sensing, which we refer to here as the Mito$_{Cat}$-Mito$_{Ox}$ model, that is relevant to both rodent and human islets and the in vivo context (*Abulizi et al., 2020*; *Lewandowski et al., 2020*; *Merrins et al., 2022*).

In the Mito$_{Cat}$-Mito$_{Ox}$ model of β-cell metabolic signaling, Ca$^{2+}$ and ADP availability dictate the metabolic cycles that preferentially occur during the triggering or secretory phases of glucose-stimulated oscillations (*Figure 1—figure supplement 1*). The triggering phase, referred to as Mito$_{Cat}$ (a.k.a. Mito$_{Synth}$; *Lewandowski et al., 2020*), is named for the matched processes of anaplerosis (i.e., the net filling of TCA cycle intermediates) and cataplerosis (i.e., the egress of TCA cycle intermediates to the cytosol). During this electrically silent phase of metabolism, the favorable bioenergetics of PEP metabolism (ΔG°=–14.8 kcal/mol for PEP vs. –7.3 for ATP) by PK progressively increases the ATP/ADP ratio, and by lowering ADP slows oxidative phosphorylation. The shift to a higher mitochondrial membrane potential (ΔΨ$_m$) elevates the NADH/NAD$^+$ ratio in the mitochondrial matrix and slows the TCA cycle, increasing acetyl-CoA that allosterically activates pyruvate carboxylase, the anaplerotic consumer of pyruvate that fuels oxaloacetate-dependent PEP synthesis by mitochondrial PEP carboxykinase (PCK2). The return of mitochondrial PEP to the cytosol completes the 'PEP cycle' that helps fuel PK, which raises ATP/ADP$_{pm}$ to close K$_{ATP}$ channels. Following membrane depolarization and Ca$^{2+}$ influx, the increased workload (ATP hydrolysis) associated with ion pumping and exocytosis elevates cytosolic ADP, which activates oxidative phosphorylation to produce ATP that sustains insulin secretion in a phase referred to as Mito$_{Ox}$. An unresolved aspect of this model is whether plasma membrane-compartmentalized PK activity is required to close K$_{ATP}$ channels. This question is important because in the current canonical model of fuel-induced insulin secretion, an increase in the bulk cytosolic ATP/ADP ratio (ATP/ADP$_c$) is generally assumed to close K$_{ATP}$ channels.

In the Mito$_{Cat}$-Mito$_{Ox}$ model, PK has two possible sources of PEP that may differentially regulate K$_{ATP}$ closure: glycolytic PEP produced by enolase, and mitochondrial PEP produced by PCK2 in response to anaplerosis (*Figure 1A*). About 40% of glucose-derived PEP is generated by PCK2 in the PEP cycle and is closely linked to insulin secretion (*Abulizi et al., 2020*; *Jesinkey et al., 2019*; *Stark et al., 2009*). However, it remains unclear how the PEP cycle influences glucose-stimulated oscillations. We hypothesize that mitochondrial PEP derived from PCK2 may provide a glycolysis-independent mechanism by which PK rapidly increases ATP/ADP$_{pm}$ locally at the K$_{ATP}$ channel in response to amino acids, which are potent anaplerotic fuels.

The isoforms of PK, each with different activities and mechanisms of control, may differentially regulate K$_{ATP}$ channels (*Figure 1A*). β-Cells express the constitutively active PKm1 as well as two allosterically recruitable isoforms, PKm2 and PKL, which are activated by glycolytic fructose-1,6-bisphosphate (FBP) generated upstream by the phosphofructokinase reaction (*DiGruccio et al., 2016*; *MacDonald and Chang, 1985*; *Mitok et al., 2018*). Pharmacologic PK activators (PKa), which lower the K$_m$ of PKm2 for PEP and increase the V$_{max}$ (*Anastasiou et al., 2012*), increase the frequency of glucose-stimulated Ca$^{2+}$ and ATP/ADP oscillations and potentiate nutrient-stimulated insulin secretion from

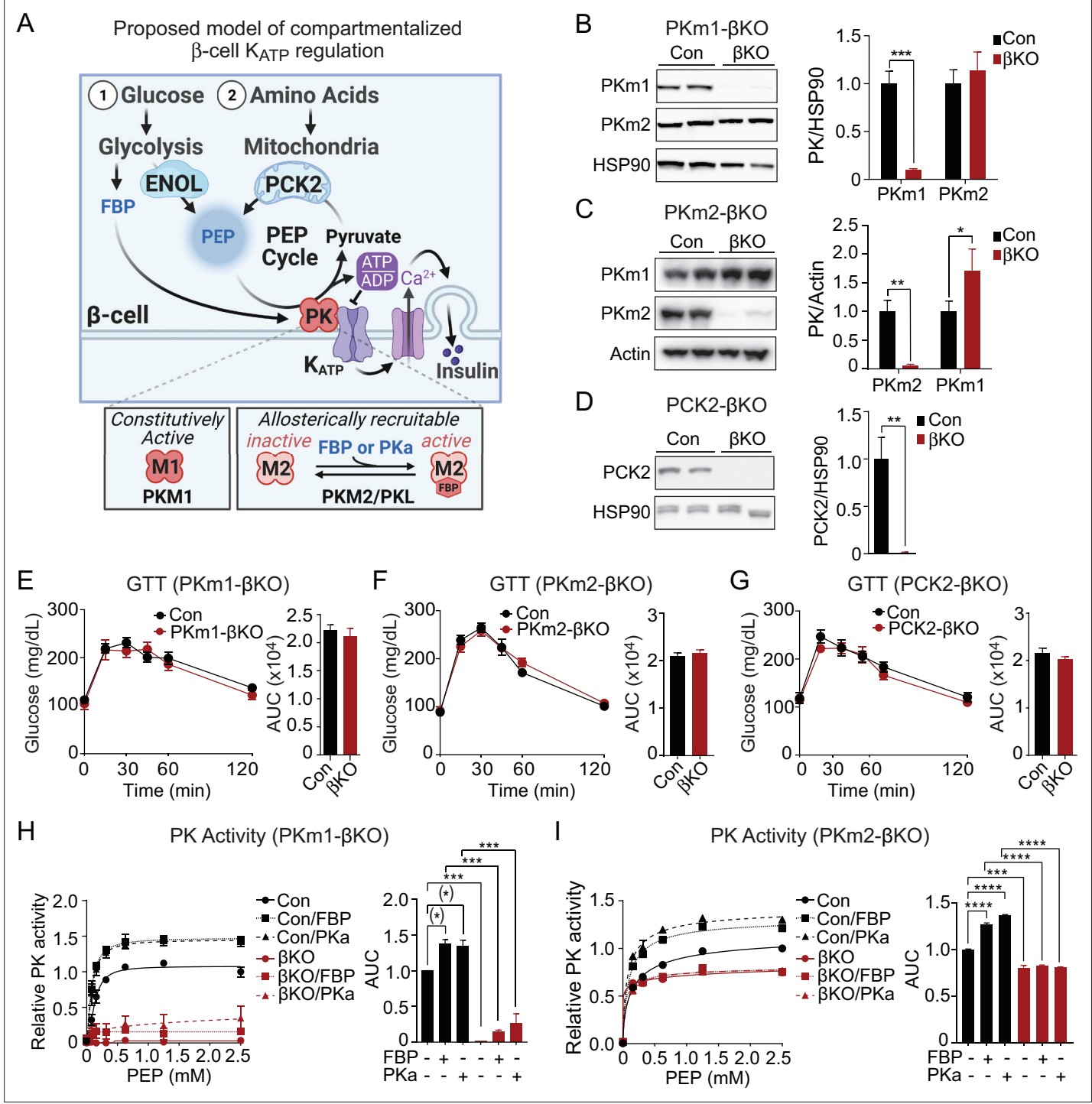

**Figure 1.** Generation of mouse models to probe the functions of PKm1, PKm2, and phosphoenolpyruvate carboxykinase (PCK2) in β-cells. (**A**) Hypothesized model in which pyruvate kinase (PK) in the K$_{ATP}$ microcompartment is fueled by two sources of phosphoenolpyruvate (PEP) – glycolytic PEP generated by enolase, and mitochondrial PEP generated by PCK2 in response to anaplerotic fuels. β-Cells express three isoforms of PK, constitutively active PKm1, and allosterically recruitable PKm2 and PKL that are activated by endogenous fructose 1,6-bisphosphate (FBP) or pharmacologic PK activators (PKa). (**B–D**) Quantification of knockdown efficiency in islet lysates from PKm1-βKO (**B**), PKm2-βKO (**C**), and Pck2-βKO mice (**D**) (n=4 mice for PKm1- and PCK2-βKO, n=6 mice for PKm2-βKO). (**E–G**) Intraperitoneal glucose tolerance tests (GTT, 1 g/kg) of PKm1-βKO mice (n=9) and littermate controls (n=8) (**E**), PKm2-βKO mice (n=7) and littermate controls (n=7) (**F**), and Pck2-βKO mice (n=10) and littermate controls (n=7) (**G**) following an overnight fast. (**H–I**) PK activity in islet lysates of PKm1-βKO (**H**) and PKm2-βKO mice (**I**) in response to FBP (80 µM) and PKa (10 µM

*Figure 1 continued on next page*

*Figure 1 continued*

TEPP-46) (n=2 replicates from 6 mice/group). See *Figure 1—source data 1* and *Figure 1—source data 2* for source data. Data are shown as mean ± SEM. #p<0.01, *p<0.05, **p<0.01, ***p<0.001, **** p<0.0001 by t-test (**B–G**) or two-way ANOVA (**H–I**).

The online version of this article includes the following source data and figure supplement(s) for figure 1:

**Source data 1.** Western blot source data.

**Source data 2.** Source data for Western blots, glucose tolerance tests, and PK activity assays associated with *Figure 1*.

**Figure supplement 1.** Cartoon depicting the $Mito_{Cat}$-$Mito_{Ox}$ model of oscillatory β-cell metabolism.

**Figure supplement 2.** Sequence verification of $PCK2^{f/f}$ mice related to *Figure 1*.

**Figure supplement 3.** Meal tolerance of PKm1-, PKm2-, and PCK2-βKO mice related to *Figure 1*.

**Figure supplement 3—source data 1.** Source data for meal tolerance tests.

rodent and human islets (*Abulizi et al., 2020*; *Lewandowski et al., 2020*). Much less is known about the PKm1 isoform, which due to its constitutive (FBP-insensitive) activity might be ideal in situations of high oxidative workload, as in cardiac myocytes (*Li et al., 2021*). β-Cells may shift their reliance upon different PK isoforms throughout the oscillatory cycle, as the levels of glycolytic FBP rise during $Mito_{Cat}$ and fall during $Mito_{Ox}$ (*Lewandowski et al., 2020*; *Merrins et al., 2016*; *Merrins et al., 2013*).

Here, we show that PK is essential for $K_{ATP}$ closure – amino acids that effectively raise $ATP/ADP_c$ cannot close $K_{ATP}$ channels without PK. We further demonstrate that both PKm1 and PKm2 are active in the $K_{ATP}$ channel microcompartment with at least two required functions. First, spatial privilege provides redundancy in the β-cell glucose response, by permitting the minor PKm2 isoform, when activated by FBP, to transmit the signal from glucose to $K_{ATP}$ despite contributing only a small fraction of the whole cell PK activity. Second, the composition of PK isoforms within the $K_{ATP}$ compartment tunes the β-cell response to amino acids, which provide mitochondrial PEP for PKm1 without also generating the FBP needed to allosterically activate PKm2. Using β-cell PCK2 deletion, we found that mitochondrially derived PEP signals to the plasma membrane PK-$K_{ATP}$ microcompartment during $Mito_{Cat}$, and facilitates $Ca^{2+}$ extrusion during $Mito_{Ox}$. These studies support the $Mito_{Cat}$-$Mito_{Ox}$ model of oscillatory metabolism, and identify unique functions of the PKm1- and PKm2-driven PEP cycles in β-cell nutrient signaling.

## Results

### β-Cell PKm1 accounts for >90% of total PK activity in mouse islets, with <10% from β-cell PKm2 and no discernable contribution from PKL

We generated β-cell-specific PKm1 and PKm2 knockout mice by breeding $Ins1^{Cre}$ mice (*Thorens et al., 2015*) with $Pkm1^{f/f}$ mice (*Davidson et al., 2021*; *Li et al., 2021*) (PKm1-βKO) or $Pkm2^{f/f}$ mice (*Israelsen et al., 2013*) (PKm2-βKO). PKm1 protein was reduced by 90% in PKm1-βKO islets, and PKm2 was not significantly increased compared to littermate $Ins1^{Cre}$ controls (*Figure 1B*). Expression of PKm2 protein fell by 94% in PKm2-βKO islets, while PKm1 increased by 29% (*Figure 1C*). This partial compensation is expected since PKm1 and PKm2 are alternative splice variants of the *Pkm* gene (*Li et al., 2021*; *Israelsen et al., 2013*). We generated $Pck2^{f/f}$ mice (*Figure 1—figure supplement 2*) and crossed them with $Ucn3^{Cre}$ mice to facilitate postnatal β-cell deletion without the need for tamoxifen (*Adams et al., 2021*; *van der Meulen et al., 2017*). Islet PCK2 protein dropped by 99% in the Pck2-βKO compared to $Pck2^{f/f}$ littermate controls (*Figure 1D*). None of these knockout mice were glucose intolerant (*Figure 1E–G*), nor did they exhibit changes in meal tolerance by oral gavage (*Figure 1—figure supplement 3A-C*).

The contributions of each PK isoform relative to total PK activity was determined in the islet lysates. The endogenous allosteric metabolite, FBP, and pharmacologic PKa such as TEPP-46 *Abulizi et al., 2020*; *Anastasiou et al., 2012*; *Lewandowski et al., 2020* have no impact on PKm1 but substantially lower the $K_m$ and raise $V_{max}$ of PKm2 or PKL (*Lewandowski et al., 2020*). Control islet lysates had a $K_m$ for PEP of 140±14 μM that was reduced in the presence of exogenous FBP ($K_m$, 100 μM±10 μM) or PKa ($K_m$, 90±13 μM), while the $V_{max}$ increased by about one-third (control, 1.05±0.019 μmol/min; FBP, 1.28±0.006 μmol/min; PKa, 1.36±0.006 μmol/min). The $V_{max}$ for islet PK activity from PKm1-βKO mice decreased by 97% compared to controls (*Figure 1H*), and was too low to estimate $K_m$ accurately.

The residual PK remained sensitive to activation by both FBP and PKa, identifying an allosterically recruitable PK pool that accounts for only about 10% of the PK activity present in control islets (*Figure 1H*). Conversely, β-cell PKm2 deletion lowered islet lysate PK $V_{max}$ by ~20% in the absence of activators, and eliminated both the $K_m$ and $V_{max}$ response to PKa and FBP (*Figure 1I*), thus ruling out any measurable PKL activity. Taken together, mouse islet PK activity is composed of >90% PKm1, with a variable contribution from PKm2 depending on the FBP level. If only considered in terms of total cellular activity related to nutrient-induced insulin secretion (i.e. in the absence of any compartmentalized functions), PKm1 should be dominant over PKm2 under all physiologic conditions. The fact that PKm1-βKO mice maintain metabolic health with unaltered glucose tolerance into adulthood suggests that the remaining PK activity is sufficient for β-cell function, and led us to hypothesize that both PKm1 and PKm2 function in the $K_{ATP}$ channel microcompartment.

## Both PKm1 and PKm2 are associated with the plasma membrane and locally direct $K_{ATP}$ channel closure, however PKm2 requires allosteric activation even at high PEP levels

We previously demonstrated that PEP, in the presence of saturating ADP concentrations, can close $K_{ATP}$ channels in mouse and human β-cells (*Lewandowski et al., 2020*). This suggests that PK is present near $K_{ATP}$ and locally lowers ADP and raises ATP to close $K_{ATP}$ channels. Excised patch-clamp experiments, which expose the inside of the plasma membrane to the bath solution (i.e. the inside-out mode), provide both the location of endogenous PK and its functional coupling to $K_{ATP}$ channels in native β-cell membranes. This approach was applied in combination with β-cell deletion of PKm1 and PKm2 to directly identify the isoforms of the enzyme present in the $K_{ATP}$ microdomain. $K_{ATP}$ channels were identified by inhibition with 1 mM ATP, which blocked the spontaneous opening that occurs after patch excision (*Figure 2A*). Channel activity was restored using a test solution containing 0.5 mM ADP and 0.1 mM ATP. In control β-cells, the further addition of 5 mM PEP closed $K_{ATP}$, as shown by a 77% reduction in the total power (a term reflecting both the frequency and channel open time) (*Figure 2A*), compared with only a 29% reduction in PKm1-βKO cells (*Figure 2B*). Note that $K_{ATP}$ channel closure occurred in control β-cells despite the continuous deluge of the channel-opener MgADP. Thus, it is not the PEP itself, but the PK activity in the $K_{ATP}$ microcompartment that is responsible for $K_{ATP}$ closure.

To test for a role of PKm2 in the $K_{ATP}$ microcompartment, PKm1-βKO cells were preincubated in the presence of 10 µM PKa, which restored PEP-dependent $K_{ATP}$ channel closure to the same extent as the control (1 mM ATP) (*Figure 2C*). PKa had a similar effect when applied acutely (*Figure 2D*), indicating that PKm2 does not require allosteric activation to localize to the plasma membrane. Although the PEP concentration is estimated to be 1 mM in rat islets (*Sugden and Ashcroft, 1977*), 5 mM PEP was chosen to exceed the $K_m$ of PKm2 in the absence of FBP (*Lewandowski et al., 2020*). Therefore, the response of PKm2 to PKa at high PEP levels indicates that $K_{ATP}$ closure requires an additional increase in the $V_{max}$ via either allosteric activation of individual subunits of PKm2, or perhaps more likely, that the functional interaction is sensitive to the quaternary structure of PKm2. β-Cells lacking PKm2 maintained channel closure with the same power as 1 mM ATP, which is attributable to the sufficiency of endogenous PKm1 (*Figure 2E*). Thus, metabolic compartmentation of PKm1 and PKm2 to the plasma membrane provides a redundant mechanism of $K_{ATP}$ channel regulation when PKm2 is allosterically activated, as well as a compelling explanation for the ability of PKm1-βKO mice to tolerate a near-complete loss of β-cell PK activity (*Figure 1E and H*).

## PKm1 and PKm2 are redundant for glucose-dependent $Ca^{2+}$ influx

The rescue of PKm1 deficiency by PKa in the $K_{ATP}$ microcompartment (*Figure 2C–D*) suggests that PKm1 and PKm2 exert shared control over $K_{ATP}$ closure, provided that glucose is present to generate FBP to activate PKm2. To test this further we examined $Ca^{2+}$ dynamics with FuraRed while using a near-infrared dye, DiR, to facilitate simultaneous imaging of PKm1-, PKm2-, and PCK2-βKO islets with their littermate controls (*Figure 3A*). β-Cell deletion of PKm1 revealed no difference in the oscillatory period or amplitude and a modest increase in the fraction of each oscillation spent in the electrically active state (i.e. the duty cycle) (*Figure 3B*). β-Cell PKm2 deletion increased the period of $Ca^{2+}$ oscillations, while having no impact on the amplitude or the duty cycle (*Figure 3D*). In addition, PKm1 and PKm2 knockouts had no discernable difference in first-phase $Ca^{2+}$ parameters (i.e. time to depolarization, amplitude, and duration of first phase) following an acute rise in glucose from 2 to 10 mM

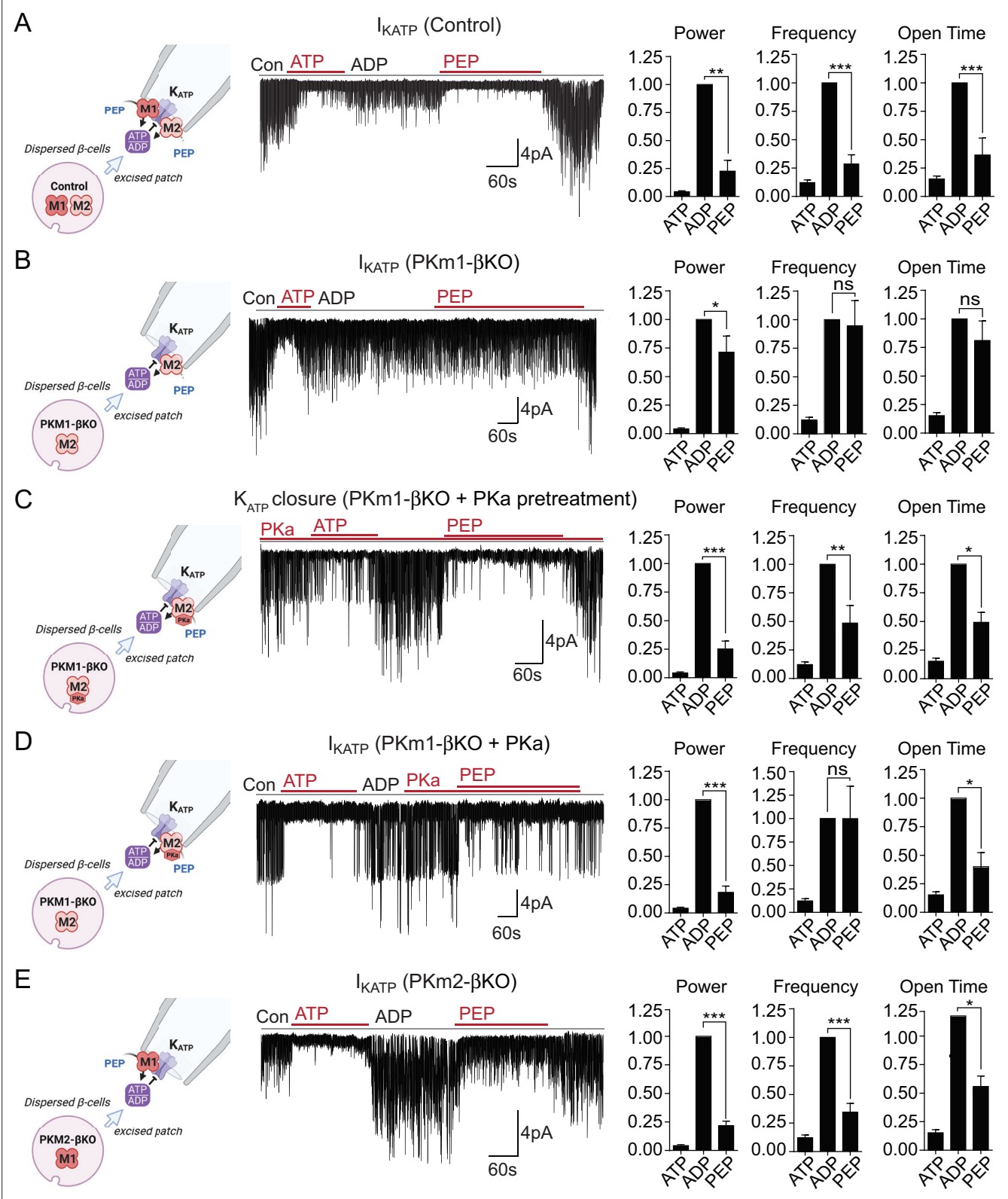

**Figure 2.** Plasma membrane $K_{ATP}$ channels are locally regulated by a combination of PKm1 and allosterically activated PKm2. (**A–E**) $K_{ATP}$ channel activity (holding potential = –50 mV) quantified in terms of power, frequency, and open time. Applying the substrates for pyruvate kinase (PK) closes $K_{ATP}$ channels in excised patches of β-cell plasma membrane from control mice (n=14 recordings from 4 mice) (**A**). Defective $K_{ATP}$ channel closure in β-cells from PKm1-KO (n=20 recordings from 5 mice) (**B**) is rescued by PK activator (PKa) pretreatment (n=6 recordings from 3 mice) (**C**) and acute PKa

*Figure 2 continued on next page*

*Figure 2 continued*

application (n=7 recordings from 3 mice) (**D**). PKm1 is sufficient for $K_{ATP}$ closure in β-cells from PKm2-βKO (n=6 recordings from 3 mice) (**E**). ATP, 1 mM; ADP, 0.5 mM ADP+ 0.1 mM ATP; PEP, 5 mM; PKa, 10 μM TEPP-46. See *Figure 2—source data 1* for source data. Data are shown as mean ± SEM. *p<0.05, **p<0.01, ***p<0.001, ****p<0.0001 by paired t-test. (*)p<0.05 by unpaired t-test in (**H**).

The online version of this article includes the following source data for figure 2:

**Source data 1.** Source data for excised patch clamp experiments in *Figure 2*.

(*Figure 3C and E*). These data confirm in situ that PKm1 and PKm2 are largely redundant at high glucose.

## PK and the PEP cycle are implicated in both on- and off-switches for Ca²⁺ influx

To study the interaction of PK with the PEP cycle, we performed islet $Ca^{2+}$ measurements using PKa and PCK2-βKO islets, in the latter case using islet barcoding to simultaneously image islets isolated from littermate controls. Consistent with the ability of allosteric PKm2 activation to accelerate $K_{ATP}$ closure, acute application of PKa to wild-type islets reduced the period as well as the amplitude of the steady-state glucose-induced $Ca^{2+}$ oscillations (*Figure 3F* and *Figure 3—figure supplement 1A*). However, we noticed that PKa shortened the time spent in $Mito_{Cat}$ and to a greater degree, $Mito_{Ox}$ (*Figure 3—figure supplement 1B*), leading to a modest reduction in the duty cycle as well as a more significant reduction in the period of the oscillation (*Figure 3F*). These observations suggest that the PKm2-driven PEP cycle regulates the onset, and even more strongly, the termination of $Ca^{2+}$ influx. Consistently, in PCK2-βKO islets where mitochondrial PEP production is inhibited, both the period and amplitude of glucose-stimulated $Ca^{2+}$ oscillations were increased relative to controls islets (*Figure 3H*). Although the duty cycle also increased, it was only by a small margin. The period lengthening occurred in part from an increased duration of $Mito_{Cat}$, and especially from an increased duration of $Mito_{Ox}$ (*Figure 3—figure supplement 1C*). Taken together, these data indicate that PKm2 controls both an 'on-switch' and an 'off-switch' for $Ca^{2+}$ oscillations, both of which depend on the mitochondrial production of PEP.

While above experiments examined conditions at a fixed elevated glucose concentration (10 mM), we also investigated $Ca^{2+}$ dynamics following the transition from low to high glucose where first-phase insulin secretion is observed. Preincubation of control islets with PKa reduced the time to depolarization as well as the duration of the first-phase $Ca^{2+}$ influx (*Figure 3G*). Conversely, depolarization was delayed in PCK2-βKO islets (*Figure 3I*). In this case, the duration of the first-phase $Ca^{2+}$ pulse was not calculated since nearly 60% of PCK2-βKO islets failed to exit the first-phase plateau in order to begin oscillations, as compared with only 27% of control islets (*Figure 3I*). In other words, while the PCK2 knockout had a weaker first-phase $Ca^{2+}$ rise, it had a much longer plateau that failed to turn off effectively. Hence, PKm2 activation and PCK2 serve as on-switches for promoting glucose-stimulated $Ca^{2+}$ influx during the triggering phase ($Mito_{Cat}$), and with a quantitatively larger effect, off-switches during the secretory phase ($Mito_{Ox}$).

## Mitochondrial PCK2 is essential for amino acids to promote a rise in cytosolic ATP/ADP

To determine whether PCK2, PKm1, or PKm2 are essential for the rise in the cytosolic ATP/ADP ratio generated by high glucose or amino acids, we used β-cell-specific expression of Perceval-HR biosensors to measure $ATP/ADP_c$. We found that, as with $Ca^{2+}$, there were no significant differences in glucose-stimulated $ATP/ADP_c$ detected in the PKm1- or PKm2-βKO islets (*Figure 4A and B*), demonstrating the redundancy of the two isoforms at high glucose for $ATP/ADP_c$ generation. Similarly, we detected no significant difference in glucose-stimulated $ATP/ADP_c$ in islets from the PCK2-βKO (*Figure 4C*).

Amino acids (AA) are obligate mitochondrial fuels that simultaneously feed oxidative and anaplerotic pathways. AA can be used as a tool for separating mechanistic components of the secretion mechanism because at low glucose they can, independently of glycolysis, raise $ATP/ADP_c$ and elicit $K_{ATP}$ channel closure, $Ca^{2+}$ influx, and insulin release. In particular, glutamine and leucine generate PEP via glutamate dehydrogenase (GDH)-mediated anaplerosis that is followed by PCK2-mediated

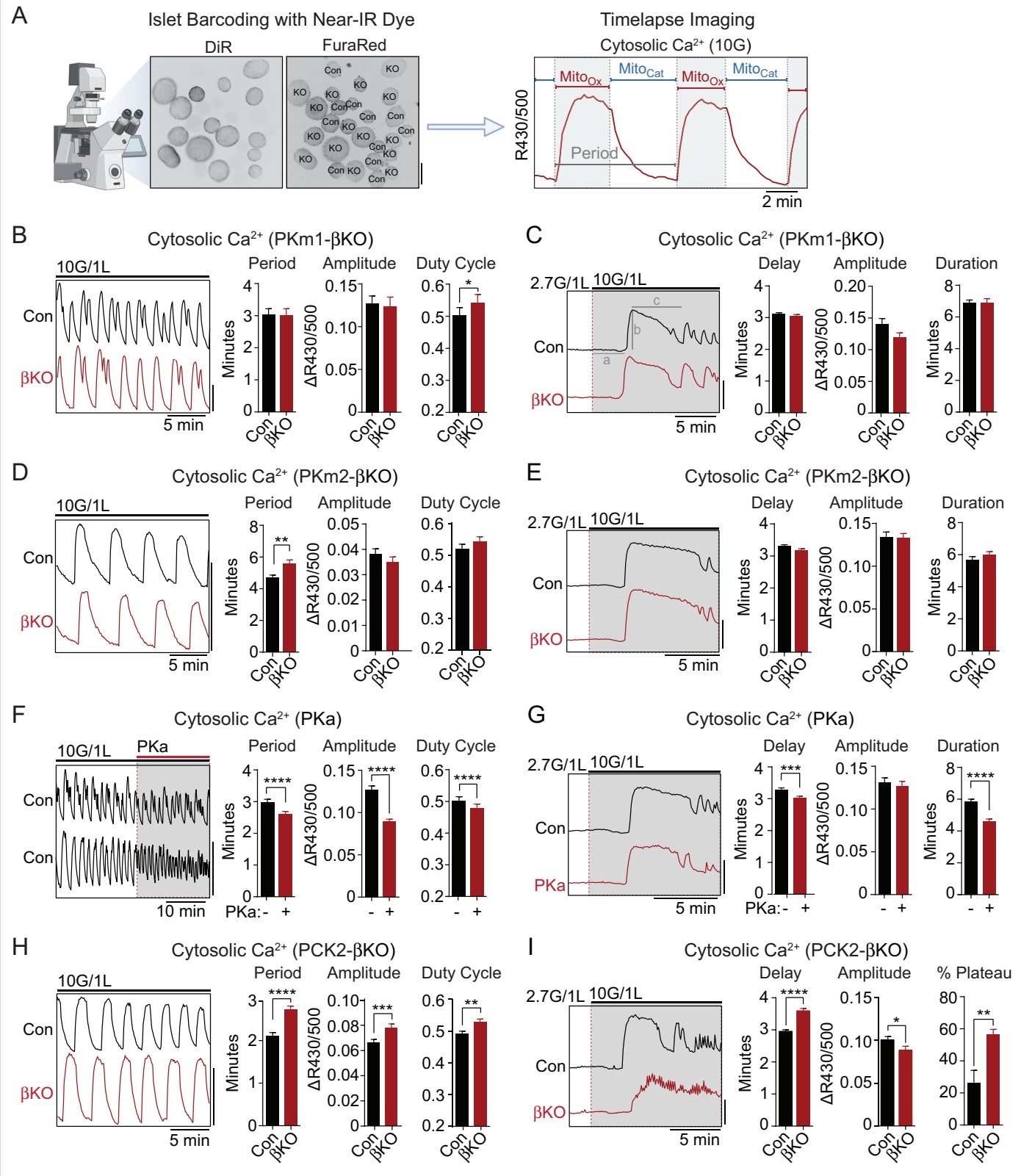

**Figure 3.** PKm2 and phosphoenolpyruvate carboxykinase (PCK2), but not PKm1, have metabolic control over first-phase and steady-state Ca²⁺ influx in response to glucose. (**A**) Barcoding of islet preparations with near-IR dye (DiR) permits simultaneous timelapse imaging of islet Ca²⁺ dynamics of control and βKO mice (scale bar = 200 µm) (left). A representative trace illustrates the cataplerotic triggering phase (Mito_Cat) and oxidative secretory phases (Mito_Ox) of steady-state Ca²⁺ oscillations in the presence of 10 mM glucose and 1 mM leucine (right). Gray line denotes the period. (**B, D, F,**

*Figure 3 continued on next page*

*Figure 3 continued*

H) Representative traces and quantification of period, amplitude, and duty cycle of steady-state Ca$^{2+}$ oscillations in islets from PKm1-βKO (n=94 islets from 3 mice) and littermate controls (n=91 islets from 3 mice) (**B**), PKm2-βKO (n=118 islets from 4 mice) and littermate controls (n=111 islets from 4 mice) (**D**), control mice treated with PK activator (PKa) (10 μM TEPP-46) (n=88 islets from 3 mice) (**F**), and PCK2-βKO (n=74 islets from 3 mice) and littermate controls (n=77 islets from 3 mice) (**H**). The bath solution (PSS) contained 10 mM glucose (10G) and 1 mM leucine. Scale bars: 0.1 FuraRed excitation ratio (R430/500). (**C, E, G, I**) Representative Ca$^{2+}$ traces and quantification of time to depolarization (**a**), first-phase amplitude (**b**), and first-phase duration (**c**) in islets from PKm1-βKO (n=144 islets from 6 mice) and littermate controls (n=150 islets from 6 mice) (**C**), PKm2-βKO (n=52 islets from 2 mice) and littermate controls (n=55 islets from 2 mice) (**E**), PKa-treated (10 μM TEPP-46) (n=161 islets from 9 mice) and vehicle controls (n=212 islets from 9 mice) (**G**), and PCK2-βKO (n=73 islets from 3 mice) and littermate controls (n=78 islets from 3 mice) (**I**). The bath solution (PSS) contained 1 mM leucine and 2.7 mM (2.7G) and 10 mM glucose (10G) as indicated. Scale bars: 0.1 FuraRed excitation ratio (R430/500). See *Figure 3—source data 1* for source data. Data are shown as mean ± SEM. *p<0.05, **p<0.01, ***p<0.001, ****p<0.0001 by unpaired t-test (**A–C, and E–I**) and paired t-test (**D**).

The online version of this article includes the following source data and figure supplement(s) for figure 3:

**Source data 1.** Source data for calcium imaging assays in *Figure 3*.

**Figure supplement 1.** Supplemental quantification of steady-state Ca$^{2+}$ dynamics related to *Figure 3*.

**Figure supplement 1—source data 1.** Source data for quantification of calcium assays in *Figure 3—figure supplement 1*.

cataplerosis of PEP (*Kibbey et al., 2014*; *Stark et al., 2009*). We first examined whether restriction of mitochondrial PEP production in PCK2-βKO islets impacts the cytosolic ATP/ADP$_c$ ratio. To limit glycolytic PEP, islets were incubated at 2.7 mM glucose. The islets were then stimulated with a mixture of AA including leucine and glutamine to allosterically activate and fuel GDH, respectively. Consistent with defective PEP cataplerosis, the ATP/ADP$_c$ response of PCK2-βKO islets was only 44% of control islets in response to AA (*Figure 4D*). In this setting of PCK2 depletion, pharmacologic PK activation did not recover any of the AA-induced ATP/ADP$_c$ response due to the absence of either a glycolytic or mitochondrial PEP source (*Figure 4G*). Comparatively, deletion of either PKm1 or PKm2 had only modest effects on the β-cell ATP/ADP$_c$ response to AA (*Figure 4E and F*). However, PKa completely recovered the AA-induced rise in ATP/ADP$_c$ in PKm1-βKO islets, in which the allosteric PKm2 isoform remains (*Figure 4H*). As expected, PK activation had no effect in PKm2-βKO islets (*Figure 4I*), confirming an on-target effect of TEPP-46.

## Mitochondrial fuels that stimulate a bulk rise in ATP/ADP$_c$ fail to close K$_{ATP}$ in the absence of PK

Mitochondria are located throughout the β-cell, including near the plasma membrane, where submembrane ATP microdomains have been observed (*Griesche et al., 2019*; *Kennedy et al., 1999*). Since PK is localized to the plasma membrane, we wondered whether during AA stimulation mitochondria can provide PEP to facilitate PK-dependent K$_{ATP}$ closure, or alternatively, whether mitochondria can serve as a direct source of ATP for K$_{ATP}$ channel closure. To determine whether mitochondrial PEP impacts the K$_{ATP}$ channel microcompartment, we monitored K$_{ATP}$ channel currents in *intact* β-cells in the cell-attached configuration in response to bath-applied AA at 2.7 mM glucose (*Figure 5A*). Mixed AA with or without PKa reduced K$_{ATP}$ channel power (reflecting the total number of transported K$^+$ ions) by ~75% in control β-cells (*Figure 5B*). However, no K$_{ATP}$ closure was observed in the absence of PCK2, even with PKa present (*Figure 5C*). These findings indicate that mitochondrially derived PEP can signal to the K$_{ATP}$ channel microcompartment, and is essential for K$_{ATP}$ closure in response to AA.

Unlike PCK2-deficient β-cells, PKm1-deficient β-cells stimulated with AA were capable of increasing ATP/ADP$_c$ to a similar level as control and PKm2-deficient β-cells (*Figure 4*). This model provides a unique opportunity to directly test the canonical model of fuel-induced insulin secretion, where a rise in ATP/ADP in the bulk cytosol is thought to be sufficient to close K$_{ATP}$ channels. While K$_{ATP}$ channels were efficiently closed by AA in control and PKm2-deficient β-cells (*Figure 5B and D*), K$_{ATP}$ channels failed to close in β-cells lacking PKm1 (*Figure 5E*). As in excised patches (*Figure 2C–D*), pharmacologic activation of PKm2 was sufficient to rescue K$_{ATP}$ closure in PKm1-deficient β-cells (*Figure 5F*). These findings demonstrate that PK activity is essential for K$_{ATP}$ channel closure in response to AA, and argue strongly against the canonical model in which mitochondrially derived ATP raises ATP/ADP$_c$ to close K$_{ATP}$ channels.

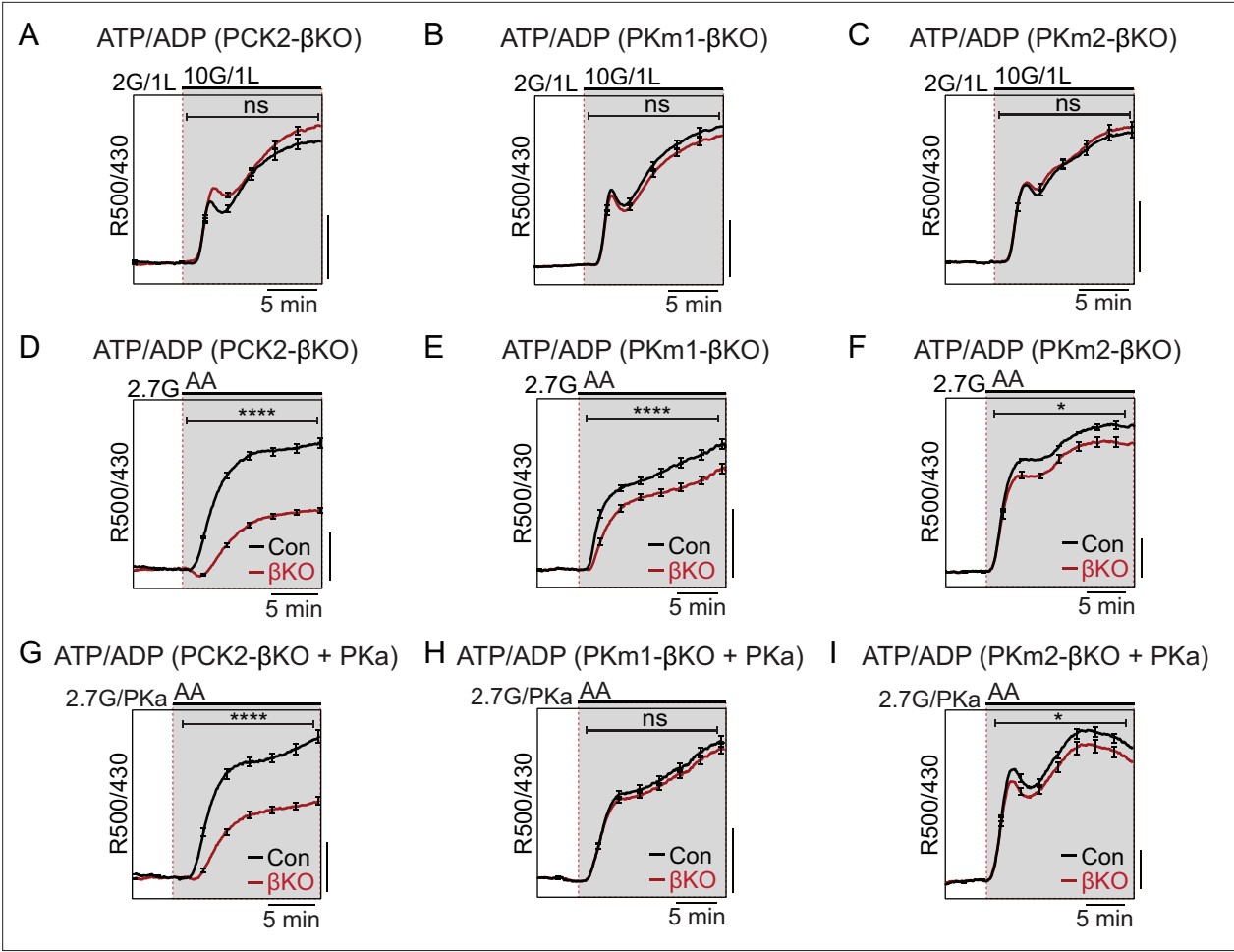

**Figure 4.** Restriction of the glycolytic phosphoenolpyruvate (PEP) supply reveals the importance of PEP carboxykinase (PCK2) for cytosolic ATP/ADP. Average β-cell ATP/ADP$_c$ in islets from PCK2-βKO (**A, D, G**), PKm1-βKO (**B, F, H**), and PKm2-βKO (**C, F, I**) mice in response to glucose in the presence of 1 mM leucine (**A–C**) or mixed amino acids (AA) provided at three times their physiological concentrations (×1=Q, 0.6 mM; L, 0.5 mM; R, 0.2 mM; A, 2.1 mM) in the presence of 2.7 mM glucose (2.7G) to remove the enolase contribution to cytosolic PEP (**D–I**). Pyruvate kinase activator (PKa) (10 μM TEPP-46) was present in G, H, and I. ATP/ADP$_c$ is quantified as area under the curve from PCK2-βKO (A, n=65 islets from 3 mice; D, n=73 islets from 3 mice; G, n=71 islets from 3 mice) and littermate controls (A, n=63 islets from 3 mice; D, n=90 islets from 3 mice; G, n=77 islets from 3 mice); PKm1-βKO (B, n=88 islets from 3 mice; E, n=66 islets from 3 mice; H, n=72 islets from 3 mice) and littermate controls (B, n=92 islets from 3 mice; E, n=69 islets from 6 mice; H, n=69 islets from 3 mice); and PKm2-βKO (C, n=86 islets from 3 mice; F, n=99 islets from 3 mice; I, n=89 islets from 3 mice) and littermate controls (C, n=90 islets from 3 mice; F, n=100 islets from 6 mice; I, n=85 islets from 3 mice). Scale bars: 0.025 Perceval-HR excitation ratio for A–C and 0.1 Perceval-HR excitation ratio for D–I (R500/430). See **Figure 4—source data 1** for source data. Data are shown as mean ± SEM. *p<0.05, ****p<0.0001 by t-test.

The online version of this article includes the following source data for figure 4:

**Source data 1.** Source data for beta cell ATP/ADP measurements in **Figure 4**.

## Mitochondrially derived PEP drives K$_{ATP}$ closure and Ca$^{2+}$ influx during Mito$_{Cat}$, and accelerates K$_{ATP}$ reopening and Ca$^{2+}$ extrusion during Mito$_{Ox}$

As for glucose, AA-stimulated Ca$^{2+}$ influx follows a distinct triggering and secretory phase, representing Mito$_{Cat}$ and Mito$_{Ox}$, respectively (**Figure 5G**). Following PK-dependent K$_{ATP}$ channel closure with AA, the ability of PKa to partially reopen K$_{ATP}$ channels (**Figure 5B**) suggests that PKm2 might also serve as an 'off-switch' during Mito$_{Ox}$ that eventually increases the channel opening by activating the PEP cycle. This concept is consistent with the ability of PKa to both hasten the onset *and* shorten the duration of Ca$^{2+}$ pulses (**Figure 3F and G**). It is also consistent with the observation that mitochondrially derived PEP is necessary to switch off glucose-dependent Ca$^{2+}$ influx, as shown in PCK2-βKO islet

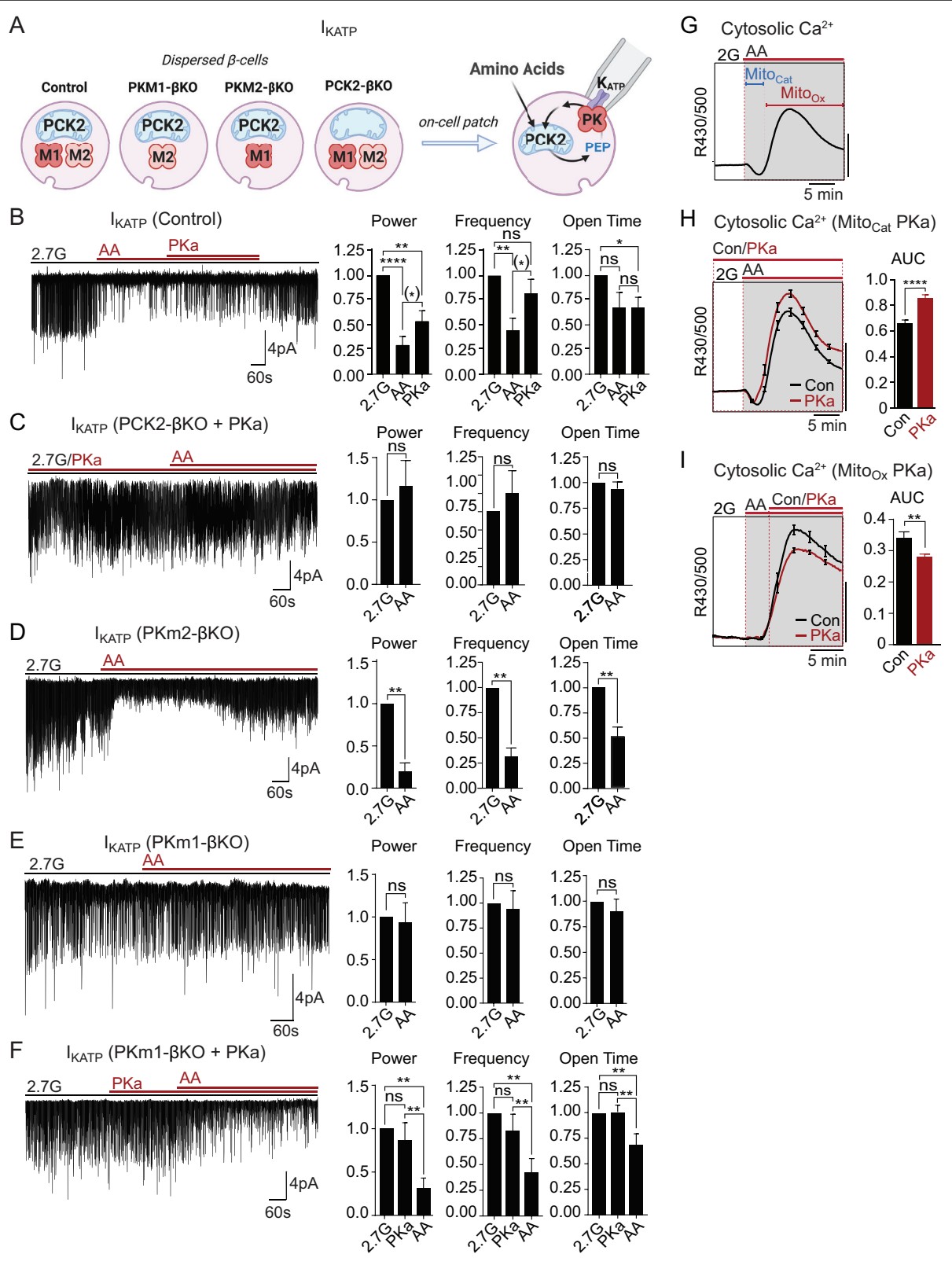

**Figure 5.** Mitochondrial phosphoenolpyruvate (PEP) signals to pyruvate kinase (PK) within the plasma membrane $K_{ATP}$ channel microcompartment in intact β-cells. (**A**) Diagram of on-cell patch clamp method in intact β-cells with bath application of amino acids (AA). (**B–F**) Representative example traces and quantification of $K_{ATP}$ channel closure in terms of normalized power, frequency, and open time for β-cells from control (n=10 recordings from 3 mice) (**B**), PCK2-βKO mice in the presence of PK activators (PKa) (10 μM TEPP-46) (n=10 recordings from 3 mice) (**C**), PKm2-βKO (n=7 recordings from

*Figure 5 continued on next page*

*Figure 5 continued*

3 mice) (**D**), and PKm1-βKO (n=7 recordings from 3 mice) (**E–F**) in response to mixed AA and 2.7 glucose (2.7G) as in *Figure 4*. (**G–I**) AA-stimulated $Ca^{2+}$ responses in control islets illustrating the $Mito_{Cat}$ and $Mito_{Ox}$ phases (**G**). The average $Ca^{2+}$ response to PKa application during $Mito_{Cat}$ (vehicle, n=575 islets from 8 mice; PKa, n=575 islets from 8 mice) (**H**) and $Mito_{Ox}$ (vehicle, n=91 islets from 4 mice; PKa, n=108 islets from 4 mice) (**I**) is quantified as AUC. Scale bars: 0.025 FuraRed excitation ratio (R430/500). See *Figure 5—source data 1* for source data. Data are shown as mean ± SEM. *p<0.05, **p<0.01, ***p<0.001, ****p<0.0001 by paired one-way ANOVA or paired t-test as appropriate. Following the removal of one outlier by ROUT (Q=10%), (*)p<0.05 by paired t-test in (**B**).

The online version of this article includes the following source data for figure 5:

**Source data 1.** Source data for on-cell patch clamp and calcium assays in *Figure 5*.

experiments (*Figure 3H and I*). In other words, the PEP cycle would have a dual function in the β-cell: during $Mito_{Cat}$, the PEP cycle facilitates $K_{ATP}$ closure and $Ca^{2+}$ influx; during $Mito_{Ox}$, the PEP cycle may facilitate $K_{ATP}$ channel reopening, $Ca^{2+}$ extrusion, and turn off insulin secretion. To test this concept further, we examined the effect of PKa on AA-stimulated $Ca^{2+}$ influx. When applied *before* AA stimulation, during $Mito_{Cat}$, PKa increased $Ca^{2+}$ influx (*Figure 5H*). By contrast, PKa application *after* the initial $Ca^{2+}$ rise, during $Mito_{Ox}$, reduced cytosolic $Ca^{2+}$ (*Figure 5I*). These data confirm two temporally separated functions of PK and mitochondrial PEP – representing on- and off-switches for β-cell $Ca^{2+}$. While the on-switch is mediated by PK-dependent closure of $K_{ATP}$ channels, the cellular location of the mitochondrial PEP-dependent off-switch is presently unclear.

## PKm1 and PKm2 respond differentially to the glycolytic and mitochondrial sources of PEP

We next examined the functional consequence of β-cell PKm1, PKm2, and PCK2 deletion on AA-stimulated $Ca^{2+}$ influx and insulin secretion at both low and high glucose (*Figure 6*). In PCK2-βKO islets at 2 mM glucose, the AA-induced $Ca^{2+}$ response was reduced along with insulin secretion (*Figure 6A and B*). In the presence of 10 mM glucose, β-cell PCK2 deletion did not impact insulin secretion because of the restored glycolytic PEP supply (*Figure 6C*). In the presence of high glucose and leucine to maximally stimulate anaplerosis, PCK2-βKO islets fail to inactivate during $Mito_{Ox}$, as indicated by the sustained $Ca^{2+}$ plateau (*Figure 3H and I*). Under similar conditions, when insulin secretion was stimulated by 10 mM glucose and mixed AA, insulin secretion was higher in PCK2-βKO islet than controls (*Figure 6C*).

Like the PCK2-βKO, the AA-stimulated $Ca^{2+}$ and secretory responses of PKm1-βKO islets were blunted at 2 mM glucose (*Figure 6D and E*). However, the insulin secretory response remained intact at 10 mM glucose in PKm1-βKO islets, whether or not AA were present (*Figure 6F*), due to the sufficiency of PKm2 in the presence of glycolytic FBP. These findings are entirely consistent with the differential response of PKm1-βKO islets to AA vs. glucose stimulation observed in the $Ca^{2+}$ and $K_{ATP}$ channel recordings above (*Figures 3 and 5*).

Similarly to PCK2, allosterically activated PKm2 has dual functions during $Mito_{Cat}$ and $Mito_{Ox}$, as evidenced by the $Ca^{2+}$ and $K_{ATP}$ channel measurements shown in *Figures 3 and 5*. Both the AA-induced $Ca^{2+}$ response and insulin secretion were greatly increased in PKm2-βKO islets compared to controls (*Figure 6G and H*). As expected, while PK activation increased AA-stimulated $Ca^{2+}$ release in control islets (*Figure 6—figure supplement 1A*), it had no effect in PKm2-βKO islets (*Figure 6—figure supplement 1B*), confirming an on-target effect of TEPP-46. Glucose alone, but especially in combination with AA, stimulated enhanced secretion in the absence of PKm2 (*Figure 6I*). Thus, while either PKm1 or PKm2 is sufficient to initiate glucose-stimulated insulin secretion during $Mito_{Cat}$, PKm2 is essential for mitochondrial PEP to switch the system off during $Mito_{Ox}$. When PKm2 was deleted, and PKm1 increased (*Figure 1C*), the system is shifted toward greater nutrient-induced and PCK2-dependent insulin secretion with heightened sensitivity to anaplerotic fuels.

## Discussion

These data provide genetic evidence that PEP has metabolic control over PK-dependent $ATP/ADP_{pm}$ generation, $K_{ATP}$ closure, $Ca^{2+}$ signaling, and insulin secretion. β-Cell PK isoform deletion experiments demonstrate that plasma membrane-associated PK is required for $K_{ATP}$ channel closure and provide rigorous genetic evidence for the $Mito_{Cat}$-$Mito_{Ox}$ model of oscillatory metabolism and insulin secretion

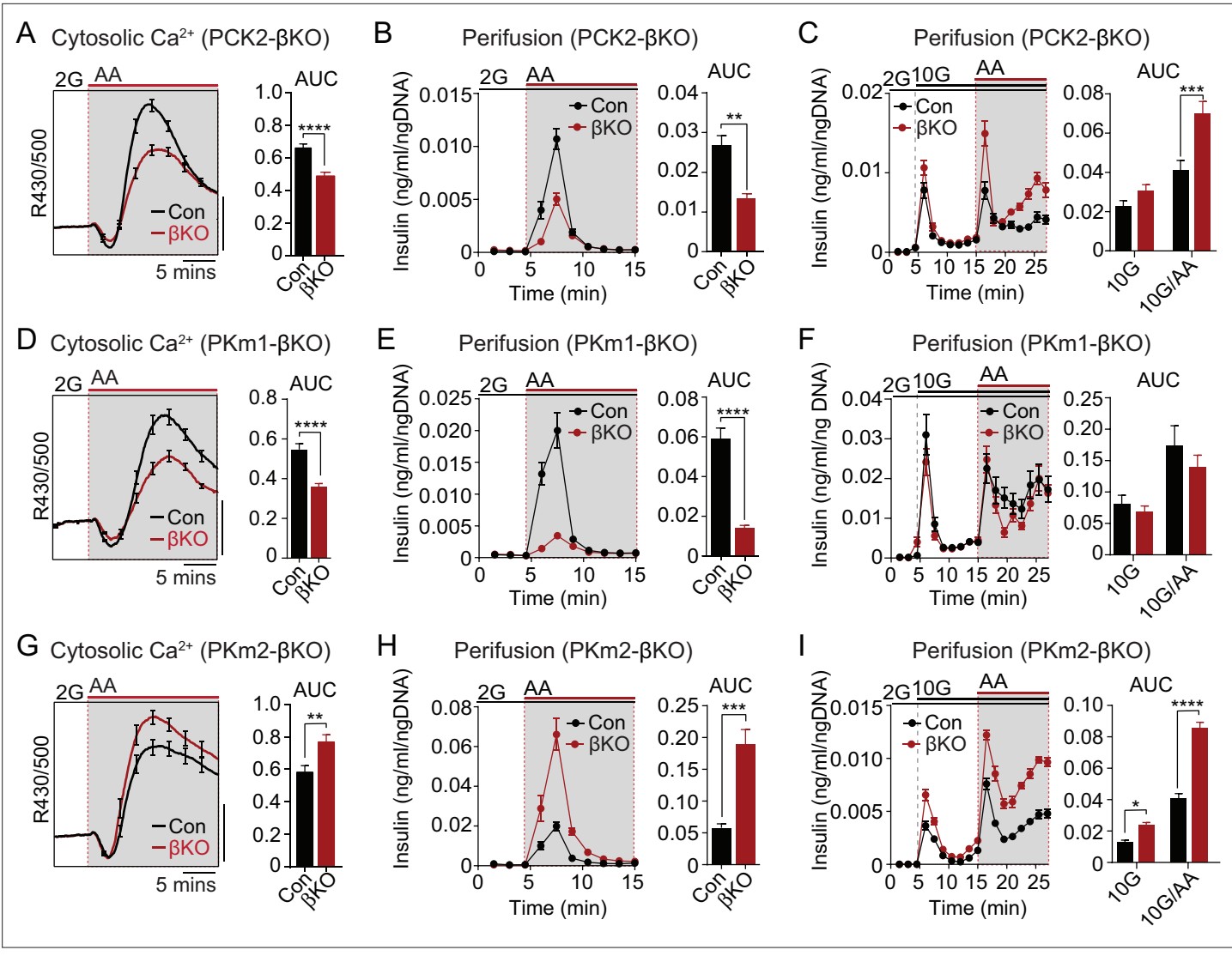

**Figure 6.** The PKm1/PKm2 ratio dictates the β-cell Ca²⁺ and secretory response to anaplerotic fuels. Average cytosolic Ca²⁺ and insulin secretory responses in controls and PCK2-βKO (**A–C**), PKm1-βKO (**D–F**), and PKm2-βKO islets (**G–I**) to the indicated concentrations of glucose (2G, 2 mM; 10G, 10 mM) and amino acids (AA; concentrations are listed in **Figure 4**). Data are quantified as AUC. Islet Ca²⁺ data reflect n=60–101 islets from 3 mice per condition (scale bar = 0.025 FuraRed excitation ratio). Islet perifusion data reflect n=100 islets per mouse and 6 mice per condition except PKm2-βKO at high glucose (**I**) (5 mice). See **Figure 6—source data 1** for source data. Data are shown as mean ± SEM. *p<0.05, **p<0.01, ***p<0.001, ****p<0.0001 by t-test. PCK2, phosphoenolpyruvate carboxykinase.

The online version of this article includes the following source data and figure supplement(s) for figure 6:

**Source data 1.** Source data for calcium and insulin secretion assays in **Figure 6**.

**Figure supplement 1.** Amino acid (AA)-stimulated Ca²⁺ influx in control and PKm2-βKO islets with and without PK activators (PKa) related to **Figure 6**.

**Figure supplement 1—source data 1.** Source data for calcium assays in **Figure 6—figure supplement 1**.

(**Merrins et al., 2022**). Our results indicate that PK is controlled by two different sources of PEP – glycolytic and mitochondrial (**Figure 1A**). Glucose signals to PK via both glycolytic and mitochondrially derived PEP, whereas AA signal to PK exclusively through mitochondrially derived PEP. Since AA do not generate FBP, which is needed to allosterically activate PKm2, PKm1 is necessary to raise ATP/ADP$_{pm}$, close K$_{ATP}$ channels, and stimulate insulin secretion in response to AA. Furthermore, our work supports the concept that the canonical model of fuel-induced insulin secretion, whereby mitochondrially derived ATP produced via the electron transport chain raises ATP/ADP$_c$ to close K$_{ATP}$ channels, is possibly wrong. The evidence for this new view of β-cell metabolic signaling is that while AA efficiently raise ATP/ADP$_c$, they do not close K$_{ATP}$ channels in the absence of plasma membrane PK activity.

Finally, the data indicate that PK and the PEP cycle have a dual role in the control of insulin secretion – they act as on-signals during the triggering phase, Mito$_{Cat}$, and as off-signals during the active secretory phase, Mito$_{Ox}$. We discuss each of these findings in the sections below.

A fuel-stimulated rise in ATP/ADP$_c$ was shown to be insufficient to close K$_{ATP}$ channels, ruling out a key aspect of the 'canonical model' in which accelerated mitochondrial metabolism raises ATP/ADP$_c$ to close K$_{ATP}$ channels (*Campbell and Newgard, 2021*; *Merrins et al., 2022*; *Prentki et al., 2013*; *Thompson and Satin, 2021*). In PKm1-deficient β-cells, AA increased ATP/ADP$_c$ similarly to control β-cells, but were unable to close K$_{ATP}$ channels. Pharmacologic activation of PKm2, present on the plasma membrane of PKm1-deficient β-cells, acutely restored K$_{ATP}$ closure. Thus, while mitochondrially derived ATP/ADP$_c$ may help to buffer the PK-dependent rise in ATPADP$_{pm}$, PK is essential for K$_{ATP}$ closure in intact β-cells. Plasma membrane-associated PK is also sufficient for K$_{ATP}$ closure in excised patch experiments, including in the presence of high concentrations of ADP that activates K$_{ATP}$ channels when complexed with Mg$^{2+}$, further supporting the concept that glycolytic ATP preferentially closes β-cell K$_{ATP}$ channels, as previously shown in cardiac myocytes (*Dhar-Chowdhury et al., 2007*; *Weiss and Lamp, 1987*). These data support the Mito$_{Cat}$-Mito$_{Ox}$ model in which plasma membrane-compartmentalized PK increases the ATP/ADP ratio near K$_{ATP}$ channels to *initiate* insulin secretion, while oxidative phosphorylation plays the dominant role in providing ATP to sustain insulin secretion *after* membrane depolarization (*Lewandowski et al., 2020*; *Merrins et al., 2022*).

In the mouse β-cell, we demonstrate that most of the PK activity is the constitutively active PKm1 isoform, with a minor contribution from PKm2 and no detectable PKL activity. In the rat, *MacDonald and Chang, 1985*, reported PK activity to be primarily PKm2, which may be a species difference. However, in the MacDonald study FBP was shown to have a minor effect on PK activity, which casts doubts about the conclusion of this work. Our prior studies of PK-dependent K$_{ATP}$ closure in human β-cell plasma membranes were conducted in the absence of FBP, which is consistent with the presence of PKm1 in the microdomain but not ruling out PKm2 (*Lewandowski et al., 2020*). Regardless, PKm2 activators amplify nutrient-stimulated insulin secretion in mouse, rat, and human islets in vitro, and in rats in vivo, indicating that PKm2 is functionally recruitable in all three species (*Abulizi et al., 2020*; *Lewandowski et al., 2020*).

Whole-body PCK2$^{-/-}$ mice display glucose intolerance and reduced insulin secretion in response to glucose and AA (*Abulizi et al., 2020*). These effects may arise independently of the β-cell, since here we show that PCK2-βKO mice of similar age are glucose tolerant, and glucose-stimulated insulin secretion from isolated islets is actually increased when AA are present. Nevertheless, Ca$^{2+}$ influx induced by AA stimulation at low glucose is blocked in both models, demonstrating the important role of mitochondrial PEP in regulating K$_{ATP}$ channels. We further demonstrate that, in response to AA, β-cell PCK2 is rate controlling for ATP/ADP$_c$, K$_{ATP}$ closure, Ca$^{2+}$ influx, and insulin secretion. However, in the presence of elevated glucose, AA stimulation in fact increased insulin secretion. This may be explained by the dual function of the PEP cycle playing a role in both the on- and off-phases of pulsatile insulin secretion. β-Cell deletion of PCK2 had a strong effect on glucose-stimulated Ca$^{2+}$ oscillations, delaying *both* the onset of Ca$^{2+}$ influx and preventing the ability of β-cells to efficiently repolarize. The failure of this off-switch provides a plausible explanation for the hypersecretion in β-cells lacking PCK2 when anaplerosis is fully primed with the combined presence of glucose and AA.

Insulin release is frequently described in terms of triggering and amplifying on-signals termed 'metabolic coupling factors' (MCFs). As defined by Prentki, 'regulatory MCFs' are nutrient-dependent signals that facilitate the switch between metabolic networks (e.g., malonyl-CoA switches β-cells from fatty acid to glucose oxidation), while 'effectory MCFs' (e.g., Ca$^{2+}$, ATP/ADP, cAMP, monoacylglycerol, and reactive oxygen species) are transient, necessary on-signals that dose-dependently stimulate insulin secretion (*Prentki et al., 2013*). Is PEP a regulatory or effectory MCF for insulin secretion – or both? PEP has some properties of a regulatory signal, since its metabolism by PK generates a bioenergetic feedforward that progressively deprives mitochondria of ADP, shutting down oxidative phosphorylation in favor of activating both pyruvate carboxylase and the PEP cycle (*Lewandowski et al., 2020*). Through a positive Hill coefficient, the increasing concentration of PEP progressively activates PKm2 to increase ATP/ADP (*Lewandowski et al., 2020*; *Merrins et al., 2013*), further reinforcing the PEP cycle. ATP/ADP$_{pm}$ is clearly an effectory signal in that it is sufficient to cause depolarization (*Ashcroft et al., 1984*; *Cook and Hales, 1984*; *Rorsman and Trube, 1985*) while at the same time priming granule exocytosis (*Eliasson et al., 1997*; *Pizarro-Delgado et al., 2016*; *Takahashi et al.,*

1999). Arguably, PEP also acts as an effectory MCF (*Sugden and Ashcroft, 1977*), since its presence in the $K_{ATP}$ microenvironment in the excised patches can override channel opening by continuous ADP to close $K_{ATP}$ channels. This property requires either PKm1 or allosterically activated PKm2.

Effectory MCFs must be counterbalanced by a strong off-switch that ensures the signal is transient. That is, the β-cell can fail if $K_{ATP}$-dependent $Ca^{2+}$ influx is activated without a coordinated homeo-static mechanism to turn it off (*Remedi and Nichols, 2009*; *Thielen and Shalev, 2018*). While PKm2 activation during $Mito_{Cat}$ was found to facilitate $K_{ATP}$ closure and increase $Ca^{2+}$, we also found that PKm2 activation during $Mito_{Ox}$ accelerated $K_{ATP}$ channel reopening and lowered $Ca^{2+}$. This off-switch most likely occurs outside the $K_{ATP}$ channel microcompartment, and may involve mitochondrial PEP, since β-cell PCK2 deletion stalls the β-cell in the $Ca^{2+}$-activated state. In addition to these temporally compartmentalized effects, spatial considerations, such as the stimulus-dependent movement of the mitochondria away from the plasma membrane (*Griesche et al., 2019*), may also be important for $Ca^{2+}$ oscillations.

Further studies are needed to elucidate precisely how the PK and the PEP cycle may contribute to turning off $Ca^{2+}$ influx and insulin secretion during the $Mito_{Ox}$ phase. Notably, these studies were limited to mice and the isoform-specific functions of PK await confirmation in human islets, although the effect of TEPP-46 to increase insulin secretion in human β-cells strongly implicates a role for PKm2/L (*Abulizi et al., 2020*; *Lewandowski et al., 2020*). Finally, it remains to be determined whether β-cell PK and the PEP cycle are altered in the states of obesity and diabetes, and whether they contribute to hyperinsulinemia and/or β-cell failure.

# Materials and methods

## Key resources table

| Reagent type (species) or resource | Designation | Source or reference | Identifiers | Additional information |
|---|---|---|---|---|
| Strain, strain background (*Mus musculus*) | B6(Cg)-*Ins1*$^{tm1.1(cre)Thor}$/J | The Jackson Laboratory | 26801 | |
| Strain, strain background (*Mus musculus*) | B6;129S-*Pkm*$^{tm1.1Mgvh}$/J | The Jackson Laboratory | 24408 | |
| Strain, strain background (*Mus musculus*) | *Pck2*$^{f/f}$ | This paper | Mutant mouse | See 'Mice' in Materials and methods. Mouse can be obtained by request from Matthew Merrins. |
| Strain, strain background (*Mus musculus*) | *Ucn3*$^{Cre}$ | Mark Huising (University of CA-Davis) and Barak Blum (University of WI-Madison) | Mutant mouse | |
| Strain, strain background (*Mus musculus*) | *Pkm1*$^{f/f}$ | Matthew Vander Heiden (MIT) | Mutant mouse | |
| Strain, strain background (*Mus musculus*) | *Pkm1*$^{f/f}$/*Ins1*$^{Cre}$ | This paper | Knockout mouse | See 'Mice' in Materials and methods. Mouse can be obtained by request from Matthew Merrins. |
| Strain, strain background (*Mus musculus*) | *Pkm2*$^{f/f}$/*Ins1*$^{Cre}$ | This paper | Knockout mouse | See 'Mice' in Materials and methods. Mouse can be obtained by request from Matthew Merrins |
| Strain, strain background (*Mus musculus*) | *Pck2*$^{f/f}$/*Ucn3*$^{Cre}$ | This paper | Knockout mouse | See 'Mice' in Materials and methods. Mouse can be obtained by request from Matthew Merrins. |
| Antibody | Anti-PKm2 (Rabbit monoclonal) | Cell Signaling | 4053 | WB (1:1000) |
| Antibody | Anti-PKm1 (Rabbit monoclonal) | Cell Signaling | 7067 | WB (1:1000) |

*Continued on next page*

*Continued*

| Reagent type (species) or resource | Designation | Source or reference | Identifiers | Additional information |
|---|---|---|---|---|
| Antibody | Anti-PCK2 (Rabbit polyclonal) | Cell Signaling | 6924 | WB (1:1000) |
| Antibody | Anti-β-Actin (Mouse monoclonal) | Cell Signaling | 3700 | WB (1:1000) |
| Antibody | Anti-HSP90 (Rabbit monoclonal) | Cell Signaling | 4877 | WB (1:1000) |
| Antibody | Anti-rabbit IgG, HRP-linked Antibody (Goat polyclonal) | Cell Signaling | 7074 | WB (1:20,000) |
| Antibody | Anti-mouse IgG, HRP-linked Antibody (Horse polyclonal) | Cell Signaling | 7076 | WB (1:20,000) |
| Chemical compound, drug | FuraRed | Molecular Probes | F3020 | |
| Chemical compound, drug | DiR'; DiIC$_{18}$(7) (1,1'-Dioctadecyl-3,3,3',3'-Tetramethylindotricarbocyanine Iodide) | Thermo Fisher Scientific | D12731 | |
| Chemical compound, drug | DiD' solid; DiIC$_{18}$(5) solid (1,1'-Dioctadecyl-3,3,3',3'-Tetramethylindodicarbocyanine, 4-Chlorobenzenesulfonate Salt) | Thermo Fisher Scientific | D7757 | |
| Other | Bio-Gel P-4 Media | Bio-Rad | 1504124 | Gio-gel used in perifusion assays |
| Commercial assay, kit | Promega Lumit Insulin Immunoassay | Promega | CS3037A01d | |
| Commercial assay, kit | PicoGreen dsDNA Assay Kit | Invitrogen | P7589 | |
| Chemical compound, drug | TEPP-46 (PKa) | MilliporeSigma Calbiochem | 50-548-70001 | |
| Chemical compound, drug | D-Fructose 1,6-bisphosphate trisodium salt hydrate | MilliporeSigma | F6803 | |
| Chemical compound, drug | Accutase solution | Sigma-Aldrich | A6964100ML | |
| Other | Microfilament borosilicate glass | Harvard Apparatus | 64-0792 | Capillaries used to make recording electrodes. |
| Software, algorithm | Clampfit analysis module | Molecular Devices | Pclamp 10 | |

## Mice

β-Cell-specific PKm1 and PKm2 knockout mice were generated by breeding *Ins1-Cre* (*Thorens et al., 2015*) mice (B6(Cg)-*Ins1*$^{tm1.1(cre)Thor}$/J, Jackson Laboratory #026801) with *Pkm1*$^{f/f}$ mice (*Davidson et al., 2021*; *Li et al., 2021*, p. 1) provided by Matthew Vander Heiden (MIT) or *Pkm2*$^{f/f}$ mice (*Israelsen et al., 2013*) (B6;129S-*Pkm*$^{tm1.1Mgvh}$/J, Jackson Laboratory #024408) after 10 generations of back-crossing to C57BL/6J mice (Jackson Laboratory). *Pck2*$^{f/f}$ mice were generated de novo by the University of Wisconsin-Madison Genome Editing and Animal Model core. Backcrossed F1s were sequence confirmed by Illumina targeted deep sequencing to confirm the LoxP insertions around exon 5 (ENSMUSE00000399990; *Figure 1—figure supplement 2*), and the intervening region was sequence confirmed with Sanger sequencing. *Pck2*$^{f/f}$ mice were crossed with *Ucn3-Cre* mice (*van der Meulen et al., 2017*) provided by Barak Blum (University of Wisconsin-Madison) with permission from Mark O Huising (University of California-Davis). Littermate *Ins1-Cre* mice were used as controls for PKm1-βKO and PKm2-βKO mice, while littermate *Pck2*$^{f/f}$ mice were used as controls for PCK2-βKO mice. Male mice were studied at 12–19 weeks of age. All mice were genotyped by Transnetyx.

## Mouse islet preparations

Male mice were housed two to five per cage at 21–23°C, fed with a chow diet and water provided ad libitum, and maintained on a 12 hr light/dark cycle. Mice 12–20 weeks of age were sacrificed via $CO_2$ asphyxiation followed by cervical dislocation, and islets were isolated using methods previously

described (*Gregg et al., 2016*). For large preparations (six or more mice), following pancreatic inflation and dissociation (*Gregg et al., 2016*), islets from each mouse were transferred to individual 50 mL conicals and washed three times with Hanks' Balanced Salt Solution (HBSS; Corning 20–023-CV) with 0.2% bovine serum albumin (Sigma-Aldrich #A9647) by centrifuging for 5 s at 800 rcf followed by removal of supernatant. Prior to the third wash, islet preparations were passed through a sieve. Following washes, islet preparations were then resuspended in 10 mL HBSS, and 10 mL of Lymphocyte Separation Medium 1077 (Sigma-Aldrich C-44010) was slowly added below the layer of islet preparation/HBSS using a 10 mL syringe with an 18 G needle (Air-Tite ML1018112). Islet preparations were centrifuged at 800 rcf for 15 min then supernatant was poured into a new 50 mL conical and HBSS/BSA was added to 50 mL. Islets were centrifuged for 5 min, supernatant was removed, and islets were resuspended in 10–20 mL, poured into a Petri dish then picked into dishes containing RPMI 1640 (Gibco #11875119).

## Glucose and meal tolerance tests
Mice aged 11–22 weeks were fasted overnight for 16 hr prior to i.p. injection of glucose (1 g/kg body weight, prepared in PBS) or oral gavage of liquid Ensure (10 mL/kg). Blood glucose was measured from the tail using a glucometer (Contour).

## Cloning and adenoviral delivery of biosensors
Generation of adenovirus carrying genetically encoded ATP/ADP biosensors (Perceval-HR) under control of the insulin promoter was described previously (*Merrins et al., 2016*). High-titer adenovirus was added to islets immediately after islet isolation and incubated for 2 hr at 37°C then moved to fresh media. Imaging was performed 3 days post isolation.

## Western blots
Islets were lysed using 0.1% Triton X-100 in phosphate buffered saline (1 μL/islet). Islets in lysis buffer were incubated at room temperature for 15 min, vortexed for 30 s, frozen/thawed, and vortexed for 30 s again. Cell lysate was spun down at max speed in a table top centrifuge and 16–20 μL supernatant was added to each well of a 12% SDS-PAGE gel. The gel was run at 110 V for ~2 hr then transferred to a PVDF membrane at 100 V for 1 hr at 4°C. Membranes were blocked in 4% BSA in Tris-buffered saline containing 0.1% Tween-20 detergent (TBST) for 30 min then incubated overnight with PKm1 (Cell Signaling #4053; 1:1000), PKm2 (Cell Signaling #7067; 1:1000), PCK2 primary antibodies (Cell Signaling #6924; 1:1000), or loading controls beta-actin (Cell Signaling #3700; 1:1000) or HSP90 (Cell Signaling #4877; 1:1000). Blots were washed for 15 min in TBST four times then incubated for 2 hr at room temperature with an anti-rabbit IgG HRP-linked secondary antibody (Cell Signaling #7074; 1:20,000) or an anti-mouse IgG HRP-linked secondary antibody (Cell Signaling #7076 1:20,000). Blots were imaged using ImageQuant (Amersham). Quantification was performed using ImageJ. To quantify both PKm1 and PKm2, blots were stripped by a 45 min incubation in stripping buffer (62.5 mM Tris pH 6.8 with 2% SDS and 0.3% β-mercaptoethanol) at 55°C, followed by three washes in TBST and overnight incubation with antibody, as described above.

## Timelapse imaging
Islets were preincubated in 2.5 μM FuraRed (Molecular Probes F3020) in RPMI 1640 for 45 min at 37°C prior to being placed in an glass-bottomed imaging chamber (Warner Instruments) on a Nikon Eclipse Ti-E inverted microscope with a Super Flour ×10/0.5 NA objective (Nikon Instruments). The chamber was perfused with standard imaging solution (in mM: 135 NaCl, 4.8 KCl, 2.6 $CaCl_2$, 20 HEPES, 1.2 $MgCl_2$) for low glucose/AA experiments, or HBSS (in mM 137 NaCl, 5.6 KCl, 1.2 $MgCl_2$, 0.5 $NaH_2PO_4$, 4.2 $NaHCO_3$, 10 HEPES, 2.6 $CaCl_2$) for high glucose experiments containing glucose and AA concentrations indicated in figure legends. Temperature was maintained at 33°C with inline solution and chamber heaters (Warner Instruments), and flow rate was set to 0.25 mL/min. Excitation was provided by a SOLA SE II 365 (Lumencor). A Hamamatsu ORCA-Flash4.0 V2 Digital CMOS camera was used to collect fluorescence emission at 0.125–0.2 Hz. Nikon NIS-Elements was used to designate regions of interest. Excitation (x) and emission (m) filters (ET type; Chroma Technology) were used with an FF444/521/608-Di01 dichroic (Semrock) as follows: FuraRed, 430/20x and 500/20x, 630/70m (R430/500 was reported) and Perceval-HR, 430/20x and 500/20x, 535/35m (R500/430 was reported).

A custom MATLAB script was used to quantify cytosolic $Ca^{2+}$ oscillation parameters (script available at https://github.com/hrfoster/Merrins-Lab-Matlab-Scripts, (**Foster, 2022** copy archived at swh:1:rev:1b-d221c947942271d1abdb4aad0e77b29cd77585)). In order to differentiate between islets of control and knockout mice, islets of one genotype were barcoded by preincubation with 2 μM DiR (Thermo Fisher Scientific D12731) or 2 μM DiD (Thermo Fisher Scientific D7757) in islet media for 10 min at 37°C. Islet barcodes were then imaged using a Cy7 (for DiR) or Cy5 filter cube (for DiD) from Chroma.

## Electrophysiology

Single channel patch clamp experiments and data analysis were performed as previously described (*Lewandowski et al., 2020*). Briefly, mouse and human cells were acutely isolated by dispersing of islets with Accutase (Sigma-Aldrich A6964100ML). Cells were plated on sterilized uncoated glass shards and kept at 37°C with 95% $O_2$-5% $CO_2$. For both on-cell and inside-out recordings, gigas-eals were obtained in extracellular bath solution (in mM): 140 NaCl, 5 KCl, 1.2 $MgCl_2$, 2.5 $CaCl_2$, 0.5 glucose, 10 HEPES, pH 7.4, adjusted with NaOH and clamped at −50 mV. For inside-out configuration, after pipette excision, the bath solution was replaced with equilibrium solutions with $K^+$ as the charge carrier (in mM): 130 KCl, 2 $CaCl_2$, 10 EGTA, 0.9 $[Mg^{2+}]_{free}$, 10 sucrose, 20 HEPES, pH 7.2 with KOH. The $[Mg^{2+}]_{free}$, calculated using WEBMAXC standard, was held constant in the presence of nucleotides. The pipette solution, used for both on-cell and inside-out configurations, contained (in mM): 10 sucrose, 130 KCl, 2 $CaCl_2$, 10 EGTA, 20 HEPES, pH 7.2, adjusted with KOH. Recording electrodes made of microfilament borosilicate glass (Harvard Apparatus, Holliston, MA, #64-0792) were used to pull patch pipettes (3 MΩ) on a horizontal Flaming/Brown micropipette puller (P-1000, Sutter Instruments) and polished by microforge (Narishige MF-830) to a final tip resistance of 5–10 MΩ. On-cell recording started after formation of a stable gigaseal (>2.5 GΩ) and inside-out recording started after withdrawal of the pipette and establishment of the excised inside-out configuration. A HEKA Instruments EPC10 patch-clamp amplifier was used for registration of current. Data was Bessel filtered online at 1 kHz and single channel currents were analyzed offline using ClampFit analysis module of pCLAMP 10 software (Molecular Devices).

## Islet perifusion assays

Islets from six mice of each genotype were pooled and then divided in equal numbers and placed into a 12-channel perifusion system (BioRep Peri-4.2 or BioRep Peri-5; 75–100 islets/chamber) containing KRPH buffer (in mM: 140 NaCl, 4.7 KCl, 1.5 $CaCl_2$, 1 $NaH_2PO_4$, 1 $MgSO_4$, 2 $NaHCO_3$, 5 HEPES, 2 glucose, 0.1% fatty acid-free BSA, pH 7.4) with 100 μL Bio-Gel P-4 Media (Bio-Rad #1504124). Islets were equilibrated at 2 mM glucose for 36–48 min prior to perfusion with AA or 10 mM glucose at 37°C. Insulin secretion was assayed using Promega Lumit Insulin Immunoassay (CS3037A01) and measured using a TECAN Spark plate reader. Quant-IT PicoGreen dsDNA Assay Kit (Invitrogen P7589) was used to determine DNA content after lysis using 0.1% Triton X-100.

## PK activity

The enzymatic activity of PK was measured using EC 2.7.1.40 from Sigma-Aldrich (*Bergmeyer et al., 1965*) as described previously (*Lewandowski et al., 2020*). FBP (Sigma-Aldrich #F6803) and TEPP-46 (PKa; MilliporeSigma Calbiochem #50-548-70001) were used at a concentration of 80 and 10 μM, respectively. Experiments were performed at 37°C.

## Quantification and statistical analysis

Figure legends describe the statistical details of each experiments. Data are expressed as mean ± SEM. Statistical significance was determined by two-way or one-way ANOVA with Sidak multiple-comparisons test post hoc or Student's t-test as indicated in the figure legends. Data were continuous and normally distributed so were analyzed with parametric tests. Differences were considered to be significant at $p < 0.05$. Calculations were performed using GraphPad Prism.

## Materials availability statement

Pck2-floxed animals will be shared on a collaborative basis pending their availability. Inquiries should be directed to the corresponding author.

## Study approval

Animal experiments were conducted under the supervision of the IACUC of the William S Middleton Memorial Veterans Hospital (Protocol: MJM0001).

## Acknowledgements

We would like to thank Matthew Vander Heiden (MIT) for providing *Pkm1*$^{f/f}$ mice, and Mark Huising (UC-Davis) and Barak Blum (UW-Madison) for providing *Ucn3*$^{Cre}$ mice. We would also like to acknowledge Kathy Krentz and Dustin Rubenstein at the UW-Madison Genome Editing and Animal Model Core for their assistance generating *Pck2*$^{f/f}$ mice, and Jiwon Seo and Jody Peter at the UW-Madison Biomedical Research Model Services Breeding Core for their assistance with animal husbandry and genotyping. Graphics were created using BioRender.com. The Merrins laboratory gratefully acknowledges support from the NIH/NIDDK (R01DK113103 and R01DK113103). This work was supported in part by the United States Department of Veterans Affairs Biomedical Laboratory Research and Development Service (I01B005113). HRF received a postdoctoral fellowships from HRSA (T32HP10010) and the NIH/NIA (T32AG000213), SLL received a predoctoral fellowship from the NIH/NIDDK (T32DK007665), HRV received a postdoctoral fellowship from the American Diabetes Association (1-17-PDF-155), and we acknowledge Dudley Lamming for contributing support for EP from R01AG062328. The Kibbey laboratory gratefully acknowledges support from the NIH/NIDDK (R01DK127637). This work utilized facilities and resources from the William S Middleton Memorial Veterans Hospital and does not represent the views of the Department of Veterans Affairs or the United States Government.

## Additional information

### Funding

| Funder | Grant reference number | Author |
| --- | --- | --- |
| National Institutes of Health | R01DK113103 | Matthew J Merrins |
| U.S. Department of Veterans Affairs | I01B005113 | Matthew J Merrins |
| Health Resources and Services Administration | T32HP10010 | Hannah R Foster |
| National Institutes of Health | T32AG000213 | Hannah R Foster |
| National Institutes of Health | T32DK007665 | Sophie L Lewandowski |
| American Diabetes Association | 1-17-PDF-155 | Halena R VanDeusen |
| National Institutes of Health | R01AG062328 | Matthew J Merrins |
| National Institutes of Health | R01DK127637 | Richard G Kibbey Matthew J Merrins |
| National Insitutes of Health | F31DK126403 | Sophie L Lewandowski |

The funders had no role in study design, data collection and interpretation, or the decision to submit the work for publication.

### Author contributions

Hannah R Foster, Data curation, Formal analysis, Investigation, Visualization, Methodology, Writing - original draft, Writing - review and editing; Thuong Ho, Evgeniy Potapenko, Sophia M Sdao, Sophie L Lewandowski, Rebecca L Cardone, Data curation, Formal analysis, Investigation; Shih Ming Huang, Formal analysis, Investigation; Halena R VanDeusen, Investigation; Shawn M Davidson, Resources; Marc Prentki, Writing - review and editing; Richard G Kibbey, Resources, Supervision, Project

administration, Writing - review and editing; Matthew J Merrins, Conceptualization, Resources, Data curation, Formal analysis, Supervision, Funding acquisition, Visualization, Methodology, Writing - original draft, Project administration, Writing - review and editing

### Author ORCIDs
Matthew J Merrins (iD) http://orcid.org/0000-0003-1599-9227

### Ethics
Animal experiments were conducted under the supervision of the IACUC of the William S. Middleton Memorial Veterans Hospital (Protocol: MJM0001).

### Decision letter and Author response
Decision letter https://doi.org/10.7554/eLife.79422.sa1
Author response https://doi.org/10.7554/eLife.79422.sa2

## Additional files

### Supplementary files
• MDAR checklist

### Data availability
Datasets generated or analyzed in this study are included in the manuscript and supporting files. Source data files are provided for Figures 1-6 and the associated figure supplement files.

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
