## [Editor Report]

This manuscript employs in vitro studies and elegant mouse models to detail how specific pyruvate kinase isoforms impact pancreatic β-cell ATP/ADP levels, ATP-sensitive K^+^ channel (KATP channel) activity, calcium handling, and insulin secretion. This is an important study that challenges the current paradigms of KATP-channel regulation, the major signaling mechanism through which pancreatic β cells couple blood glucose levels to insulin release. Future studies will determine whether similar mechanisms are used in human pancreatic β cells.

---

## [Decision Letter]

**Decision letter after peer review:**

Thank you for submitting your article "The isoforms of pyruvate kinase act as nutrient sensors for the β-cell KATP channel" for consideration by *eLife*. Your article has been reviewed by 3 peer reviewers, and the evaluation has been overseen by a Reviewing Editor and David James as the Senior Editor. The following individual involved in review of your submission has agreed to reveal their identity: David J Hodson (Reviewer #2): Melkam Kebede (Reviewer #3).

The reviewers have discussed their reviews with one another and are uniformly positive about your manuscript and its valuable contribution to the field. The Reviewing Editor has drafted this to help you prepare a revised submission.

Essential revisions:

(1) Experimentally address reviewer's 1 concern about potential off-target effects of TEPP-46, especially with regards to the potential effect of TEPP-46 on calcium handling.

(2) Use at least one of the mouse KO models to determine whether there is a response in a mixed meal tolerance test that would correspond to the observed altered response to amino acids in vitro. Alternatively, it would be essential to provide additional discussion of the discordance between the in vitro response to amino acids and in vivo experiments showing a lack of response to glucose.

(3) Add discussion of why only male animals were used in the study and how this contributes to limitations of the study.

(4) Add discussion of the caveats of using only mouse models and how these findings could potentially translate (or not) to humans. The word "mice" should be added to the title.

(5) The reviewers provided additional constructive suggestions to improve the manuscript. Please address these experimentally when feasible or provide additional information that will address/mitigate these concerns.

*Reviewer #1 (Recommendations for the authors):*

– The authors have clearly shown that only PKm2 activity is enhanced by the PK activator TEPP-46 PMID: 33147484; "TEPP-46 (PKa), have no impact on PKm1 but substantially lower the Km and raise Vmax of PKm2" and that this is lost in islets with β-cell ablation of specific of PKm2 (Figure 1I showing no PK activation by TEPP-46 in PKm2-βKO islets). This strongly suggests that the increase in islet calcium oscillations shown when stimulated with TEPP-46 (Figure 3F) are likely to be mediated via PKm2 activation. It would be important to confirm this by assessing the impact of TEPP-46 on PKm2-βKO islet calcium oscillations in 10 mM glucose.

– Although the authors show that glucose-induced islet calcium entry is greater in terms of amplitude and oscillation duration, 2nd phase insulin secretion from PCK2-βKO islets does not seem to be impacted. The manuscript does show that in the presence of high glucose that amino acids immediately increase insulin secretion in PCK2-βKO islets above controls (figure 6c). However, the amplitude of islet calcium in response to glucose and amino acids is reduced PCK2-βKO islets (figure 6b). Some experiment showing that under the same conditions of insulin perifusion (figure 6c) that islet calcium shows elevation with AA stimulation following glucose pretreatment would be useful.

– The authors mention that "following membrane depolarization, the PEP cycle is also involved in an off-switch that facilitates KATP channel reopening and Ca^2+^ extrusion, as shown by PK activation experiments and β-cell PCK2 deletion that prolonged Ca^2+^ oscillations and increased insulin secretion." The "off-switch" is proposed to be mediated via KATP activation, and it is not clear how the factors that the authors have shown to block KATP channels are so tightly coupled with KATP activation. PK would be predicted to produce ATP close to KATP channels, but not to consume ATP close to these channels; thus, the authors should indicate how PEP (that has been shown to inhibit KATP) is linked to ATP utilization in a microdomain close to the KATP channels. The data do show that following KATP channel closure with amino acids that PKa can partially reopen KATP channels within a minute. While MitoOx could lead to eventual ATP reduction, the TEPP-46 should still activate PK close to KATP channels (especially immediately after addition in attached patch confirmation), which would locally increase ATP near the channels and close them further (this does not seem to be observed in the single channel activity show in Figure 5b).

Moreover, the authors show that TEPP-46 addition during AA induced calcium entry reduced calcium entry likely through activation of KATP channels (yet PK activation near the KATP channels would still be predicted to transfer phosphate from phosphoenolpyruvate (PEP) to ADP to produce ATP leading to further KATP closure); it is not clear how TEPP-46 addition during AA induced calcium entry would so quickly consume ATP next to the KATP channel (especially because PKm2-βKO islets show reduced ATP/ADP in response to amino acid stimulation, which suggests that TEPP-46 addition would actually increase islet ATP/ADP). The authors indicate that MitoOx is initiated during amino-acid induced calcium influx (figure 5G). While figure 5b suggests that ATP turnover (also shown in Supplemental Figure S1) occurs during amino acid induced islet calcium influx, in figure 4 the Percival recordings show an increase in the ATP/ADP ratio during amino acid stimulation when islet calcium would be high. The increase in the islet ATP/ADP ratio in response to amino acid stimulation of islet calcium entry (Figure 5) differs from what happens during an islet glucose-stimulated calcium oscillation where ATP/ADP drops during the elevation in islet calcium (PMID: 33147484); this suggests that the amino acid experiments are not showing ATP turnover that the authors indicate happens during MitoOx (Figure 3A and PMID: 33147484; also Supplemental Figure S1). The Percival recordings in response to amino acids are monitoring cytoplasmic ATP/ADP that reflects mitochondrial activity and importantly loss of β-cell PKm2 reduces the ATP/ADP ratio stimulated by amino acids compared to controls. Thus, it is important to define how global ATP could be rising in response to amino acid stimulation, but KATP channels are instead activated by TEPP-46; this is important because TEPP-46 activation of PKm2would be predicted to increase global islet ATP/ADP in response to amino acid stimulation, which is predicted from the PKm2-βKO islets showing reduced ATP/ADP in response to amino acid stimulation (Figure 4E). Taken together, the authors should define how PK activation induces specific ATP depletion near the KATP channel and not in the cytoplasm during amino acid stimulation. Also, more clarification on PK-mediated KATP activation is required in particular in relation to the Percival data of this manuscript and differences in ATP/ADP ratio in the amino acid MitoOx conditions (with increased ATP/ADP) compared to glucose-stimulated calcium oscillation MitoOx conditions (with decreased ATP/ADP).

*Reviewer #2 (Recommendations for the authors):*

Please find below a number of constructive comments to help strengthen the conclusions of the study:

– The title should be modified to include 'in mice', since the results in their present form are only applicable to mouse β cells. This is not a problem per se, since mice are the most widely used model for investigating KATP channel activity, and the genetic manipulations used here cannot be easily performed in human islets.

– "β-cell deletion of PKm1 or PKm2 did not reveal any biologically meaningful differences in the oscillatory period, the fraction of each oscillation spent in the electrically-active state (i.e. the duty cycle), or the amplitude of glucose-stimulated Ca^2+^ oscillations in intact islets (Figure 3b, d)." Please qualify how 'biologically meaningful' was assessed. The increase in Ca^2+^ oscillation period in Figure 3D (Pkm2-BKO) is the same magnitude as the changes reported in Figure 3F and G (PKa), which are stated to be 'reduced'.

– In experiments reporting G2.7 -> G10, why does it take ~3-5 mins to observe a rise in intracellular Ca^2+^? Does the grey box indicate G10 entering the perifusion system, or achievement of G10 in the cuvette?

– "Thus, while neither PK isoform is required for AA stimulation to increase ATP/ADPc, PCK2 displays high control strength over ATP/ADPc in response to AA, especially considering that the PCK2 protein was only reduced by about two-thirds (Figure 1d)." Logic of this sentence is unclear. Both PKm1 and PKm2 KO impair ATP/ADP responses to AA. So the kinases are required for AAs to increase ATP/ADP to maximal. In any case, even PCK2-BKO does not prevent AAs from stimulating ATP/ADP.

– "These remarkable findings", "novel" etc. Please remove statements of novelty. What one individual finds remarkable, another might find unsurprising.

– While not essential, this reviewer found themselves wanting Ca^2+^ recordings to confirm that changes in KATP-channel activity are linked to downstream Ca^2+^ fluxes and alterations in insulin during AA + high glucose (G10) perifusion insulin assays (Figure 6).

– Maybe I failed to make a link somewhere, but how do impaired cytoplasmic ATP/ADP rises in PKm2-BKO islets lead to apparent increases in intracellular Ca^2+^/insulin secretion?

– The majority of the data are from in vitro models, with the various KO mice showing no changes in glucose homeostasis. However, data in Figure 6C, F suggest that PKm1 and PKm2mice should respond to a mixed meal tolerance tests (AA + glucose) with decreases in blood glucose (or second phase insulin release). This would be a powerful corroborative link between in vitro and in vivo phenotype unless the authors believe this not to be the case.

– "Since amino acids do not generate FBP, which is needed to allosterically activate PKm2, PKm1 is necessary to raise ATP/ADPpm, close KATP channels, and stimulate insulin secretion in response to amino acids." I don't recall that the authors measured plasma membrane (PM)-localised ATP/ADP rises. A PM targeting sequence might be something to add to Perceval-HR in the future.

– Male mice are used throughout the study. Either the authors should confirm major findings in female mice, or discuss this limitation in the text.

– Please show full unedited Western blots in the Supplementary Information, as well as describe the loading controls e.g. was this the same lanes, parallel lanes or a separate blot?

*Reviewer #3 (Recommendations for the authors):*

In Figures 1B and 1D the western images do not represent the quantification bar graphs. PKm1 seems completely absent in 1B and same for Pkm2 in 1D.

---

## [Author Response]

Essential revisions:(1) Experimentally address reviewer's 1 concern about potential off-target effects of TEPP-46, especially with regards to the potential effect of TEPP-46 on calcium handling.

We agree it is important to evaluate TEPP-46 in the context of the PKm2-βKO, and have now performed two independent sets of experiments to demonstrate that TEPP-46 is on target. First is that TEPP-46 increases amino acid-stimulated Ca^2+^ influx in control islets, while having no effect in PKm2-βKO islets. These data are now shown in Figure 6—figure supplement 1. Second, TEPP-46 rescued the defect in amino acid-stimulated ATP/ADP_c_ rise observed in β-cells from PKm1-βKO islets, while having no effect in PKm2-βKO islets. These data are shown in Figure 4i. These two assays demonstrate that PKm2 mediates the effect of TEPP-46 on cytosolic Ca^2+^ and ATP/ADP.

(2) Use at least one of the mouse KO models to determine whether there is a response in a mixed meal tolerance test that would correspond to the observed altered response to amino acids in vitro. Alternatively, it would be essential to provide additional discussion of the discordance between the in vitro response to amino acids and in vivo experiments showing a lack of response to glucose.

We have now included MTTs for all three mouse models. Like the GTTs shown in Figure 1, there are no differences in meal tolerance in the PKm1-βKO, PKm2-βKO, and PCK2-βKO mice in comparison to controls (Figure 1—figure supplement 3). Many β-cell KOs exhibit changes in insulin secretion that are not reflected by a change in glucose tolerance due to compensations that occur over time in vivo. Thus, we don’t view the perifusion data as discordant with the GTT/MTT.

(3) Add discussion of why only male animals were used in the study and how this contributes to limitations of the study.

A section on limitations is now included in the last paragraph of the discussion.

(4) Add discussion of the caveats of using only mouse models and how these findings could potentially translate (or not) to humans. The word "mice" should be added to the title.

We have resubmitted with a new title: “β-cell deletion of the PKm1 and PKm2 isoforms of pyruvate kinase in mice reveals their essential role as nutrient sensors for the KATP channel.” In the discussion, we note that PK is sufficient to close KATP channels in human β-cells, and that PK activators targeting PKm2/L increase insulin secretion in human islets.

(5) The reviewers provided additional constructive suggestions to improve the manuscript. Please address these experimentally when feasible or provide additional information that will address/mitigate these concerns.

We appreciate the reviewers’ feedback and have addressed their comments below.

Reviewer #1 (Recommendations for the authors):– The authors have clearly shown that only PKm2 activity is enhanced by the PK activator TEPP-46 (PMID: 33147484; "TEPP-46 (PKa)), have no impact on PKm1 but substantially lower the Km and raise Vmax of PKm2" and that this is lost in islets with β-cell ablation of specific of PKm2 (Figure 1I showing no PK activation by TEPP-46 in PKm2-βKO islets). This strongly suggests that the increase in islet calcium oscillations shown when stimulated with TEPP-46 (Figure 3F) are likely to be mediated via PKm2 activation. It would be important to confirm this by assessing the impact of TEPP-46 on PKm2-βKO islet calcium oscillations in 10 mM glucose.

We agree it is important to evaluate TEPP-46 in the context of the PKm2-βKO, and have now performed two independent sets of experiments to demonstrate that TEPP-46 is on target and works via PKm2. First is that TEPP-46 increases amino acid-stimulated Ca^2+^ influx in control islets, while having no effect in PKm2-βKO islets. These data are now shown in Figure 6—figure supplement 1. Second, TEPP-46 rescued the defect in amino acid-stimulated ATP/ADP_c_ rise observed in β-cells from PKm1-βKO islets, while having no effect on β-cell ATP/ADP_c_ in PKm2-βKO islets. These data are shown in Figure 4i. These assays demonstrate that PKm2 mediates the effect of TEPP-46 on cytosolic Ca^2+^ and ATP/ADP.

– Although the authors show that glucose-induced islet calcium entry is greater in terms of amplitude and oscillation duration, 2nd phase insulin secretion from PCK2-βKO islets does not seem to be impacted. The manuscript does show that in the presence of high glucose that amino acids immediately increase insulin secretion in PCK2-βKO islets above controls (figure 6c). However, the amplitude of islet calcium in response to glucose and amino acids is reduced PCK2-βKO islets (figure 6b). Some experiment showing that under the same conditions of insulin perifusion (figure 6c) that islet calcium shows elevation with AA stimulation following glucose pretreatment would be useful.

The reviewer may have missed that the islet Ca^2+^ measurements in Figure 3 were done in the presence of high glucose and 1 mM leucine to maximally stimulate anaplerosis. Although this was previously indicated in the text and figure legends, we have now denoted the presence of leucine in Figure 3 itself. Thus, the Ca^2+^ phenotype in PCK2-βKO islets (Figure 3h, i) is well-matched to the insulin secretion (Figure 6c). Notably, Ca^2+^ is plateaued under the glucose + mixed AA condition shown in Figure 6, which is why we used glucose + leucine in Figure 3.

– The authors mention that "following membrane depolarization, the PEP cycle is also involved in an off-switch that facilitates KATP channel reopening and Ca^2+^ extrusion, as shown by PK activation experiments and β-cell PCK2 deletion that prolonged Ca^2+^ oscillations and increased insulin secretion." The "off-switch" is proposed to be mediated via KATP activation, and it is not clear how the factors that the authors have shown to block KATP channels are so tightly coupled with KATP activation. PK would be predicted to produce ATP close to KATP channels, but not to consume ATP close to these channels; thus, the authors should indicate how PEP (that has been shown to inhibit KATP) is linked to ATP utilization in a microdomain close to the KATP channels. The data do show that following KATP channel closure with amino acids that PKa can partially reopen KATP channels within a minute. While MitoOx could lead to eventual ATP reduction, the TEPP-46 should still activate PK close to KATP channels (especially immediately after addition in attached patch confirmation), which would locally increase ATP near the channels and close them further (this does not seem to be observed in the single channel activity show in Figure 5b).Moreover, the authors show that TEPP-46 addition during AA induced calcium entry reduced calcium entry likely through activation of KATP channels (yet PK activation near the KATP channels would still be predicted to transfer phosphate from phosphoenolpyruvate (PEP) to ADP to produce ATP leading to further KATP closure); it is not clear how TEPP-46 addition during AA induced calcium entry would so quickly consume ATP next to the KATP channel (especially because PKm2-βKO islets show reduced ATP/ADP in response to amino acid stimulation, which suggests that TEPP-46 addition would actually increase islet ATP/ADP). The authors indicate that MitoOx is initiated during amino-acid induced calcium influx (figure 5G). While figure 5b suggests that ATP turnover (also shown in Supplemental Figure S1) occurs during amino acid induced islet calcium influx, in figure 4 the Percival recordings show an increase in the ATP/ADP ratio during amino acid stimulation when islet calcium would be high. The increase in the islet ATP/ADP ratio in response to amino acid stimulation of islet calcium entry (Figure 5) differs from what happens during an islet glucose-stimulated calcium oscillation where ATP/ADP drops during the elevation in islet calcium (PMID: 33147484); this suggests that the amino acid experiments are not showing ATP turnover that the authors indicate happens during MitoOx (Figure 3A and PMID: 33147484; also Supplemental Figure S1). The Percival recordings in response to amino acids are monitoring cytoplasmic ATP/ADP that reflects mitochondrial activity and importantly loss of β-cell PKm2 reduces the ATP/ADP ratio stimulated by amino acids compared to controls. Thus, it is important to define how global ATP could be rising in response to amino acid stimulation, but KATP channels are instead activated by TEPP-46; this is important because TEPP-46 activation of PKm2would be predicted to increase global islet ATP/ADP in response to amino acid stimulation, which is predicted from the PKm2-βKO islets showing reduced ATP/ADP in response to amino acid stimulation (Figure 4E). Taken together, the authors should define how PK activation induces specific ATP depletion near the KATP channel and not in the cytoplasm during amino acid stimulation. Also, more clarification on PK-mediated KATP activation is required in particular in relation to the Percival data of this manuscript and differences in ATP/ADP ratio in the amino acid MitoOx conditions (with increased ATP/ADP) compared to glucose-stimulated calcium oscillation MitoOx conditions (with decreased ATP/ADP).

This seems to be a simple misunderstanding. In agreement with the reviewer, we do not believe that the off-switch is mediated by KATP channel reactivation by PK (see the text above). Although the application of TEPP-46 during Mito_Ox_ accelerates KATP channel reopening and reduces Ca^2+^ (i.e. an “off-switch”; Figure 5b, i), the PK reaction is effectively irreversible so there is no way that the local consumption of ATP by PK in the KATP channel microdomain mediates this effect. In the revised text, we clearly state that the off-switch functions only after KATP channel closure and most likely involves a multi-step process outside of the KATP channel microdomain. Indeed, our findings suggest that the off-switch requires the mitochondrial PEP cycle, since PCK2-βKO islets exhibited prolonged Ca^2+^ influx and fail to efficiently repolarize.

Reviewer #2 (Recommendations for the authors):Please find below a number of constructive comments to help strengthen the conclusions of the study:– The title should be modified to include 'in mice', since the results in their present form are only applicable to mouse β cells. This is not a problem per se, since mice are the most widely used model for investigating KATP channel activity, and the genetic manipulations used here cannot be easily performed in human islets.

We have resubmitted with a new title: “β-cell deletion of the PKm1 and PKm2 isoforms of pyruvate kinase in mice reveal their essential role as nutrient sensors for the KATP channel.”

– "β-cell deletion of PKm1 or PKm2 did not reveal any biologically meaningful differences in the oscillatory period, the fraction of each oscillation spent in the electrically-active state (i.e. the duty cycle), or the amplitude of glucose-stimulated Ca^2+^ oscillations in intact islets (Figure 3b, d)." Please qualify how 'biologically meaningful' was assessed. The increase in Ca^2+^ oscillation period in Figure 3D (Pkm2-BKO) is the same magnitude as the changes reported in Figure 3F and G (PKa), which are stated to be 'reduced'.

This was an error, and we thank the reviewer for pointing it out. We have reported the increased period in PKm2-βKO islets and the increased duty cycle in the PKm1-βKO islets as statistically significant differences.

– In experiments reporting G2.7 -> G10, why does it take ~3-5 mins to observe a rise in intracellular Ca^2+^? Does the grey box indicate G10 entering the perifusion system, or achievement of G10 in the cuvette?

The grey box indicates the time when glucose reaches the islets. Since KATP channel closure occurs quickly, we do not know why it takes 3-5 minutes for membrane depolarization, but this timing has been consistently observed by many studies and no satisfying explanation has emerged in the literature. Given that each β-cell is estimated to be coupled to 6 other β-cells by gap junctions, we speculate that the islet behavior reflects the need for first responder cells to overcome the inhibitory conductance from gap junctions. Outside of reporting that PK and the PEP cycle accelerate the time to depolarization (Figure 3), the network behavior of the islet is beyond the scope of this paper.

– "Thus, while neither PK isoform is required for AA stimulation to increase ATP/ADPc, PCK2 displays high control strength over ATP/ADPc in response to AA, especially considering that the PCK2 protein was only reduced by about two-thirds (Figure 1d)." Logic of this sentence is unclear. Both PKm1 and PKm2 KO impair ATP/ADP responses to AA. So the kinases are required for AAs to increase ATP/ADP to maximal. In any case, even PCK2-BKO does not prevent AAs from stimulating ATP/ADP.

We agree with the reviewer and have removed this sentence.

– "These remarkable findings", "novel" etc. Please remove statements of novelty. What one individual finds remarkable, another might find unsurprising.

Thank you for this comment. We have removed statements of novelty from the manuscript.

– While not essential, this reviewer found themselves wanting Ca^2+^ recordings to confirm that changes in KATP-channel activity are linked to downstream Ca^2+^ fluxes and alterations in insulin during AA + high glucose (G10) perifusion insulin assays (Figure 6).

Reviewer 1 also brought this up. We apologize if it was unclear that the islet Ca^2+^ measurements in Figure 3 were done in the presence of high glucose and 1 mM leucine to maximally stimulate anaplerosis. Although this was indicated in the text and figure legends, we have now denoted the presence of leucine in Figure 3 itself. Thus, the Ca^2+^ phenotype in PCK2-βKO islets (Figure 3h, i) is well-matched to the insulin secretion (Figure 6c). Notably, ca^2+^ is plateaued under the glucose + mixed AA condition shown in Figure 6, which is why we used glucose + leucine in Figure 3.

– Maybe I failed to make a link somewhere, but how do impaired cytoplasmic ATP/ADP rises in PKm2-BKO islets lead to apparent increases in intracellular Ca^2+^/insulin secretion?

The short answer is we don’t know. As suggested in the results and discussion, the increased cytosolic Ca^2+^ and insulin secretion observed in the β-cell PKm2 knockout may be due to the loss of the off-switch that we discovered. The cellular location and molecular mechanism of this off-switch remains to be elucidated, although it does appear to involve the mitochondrial PEP cycle, since β-cell PCK2 deletion stalls islets in the Ca^2+^ activated state.

– The majority of the data are from in vitro models, with the various KO mice showing no changes in glucose homeostasis. However, data in Figure 6C, F suggest that PKm1 and PKm2mice should respond to a mixed meal tolerance tests (AA + glucose) with decreases in blood glucose (or second phase insulin release). This would be a powerful corroborative link between in vitro and in vivo phenotype unless the authors believe this not to be the case.

This was a good suggestion and we performed MTTs. However, like the GTTs shown in Figure 1, there are no differences in meal tolerance in the PKm1-βKO, PKm2-βKO, and PCK2-βKO mice in comparison to controls. These data are now shown in Figure 1—figure supplement 3. We are not particularly surprised by these data, since many β-cell knockouts exhibit changes in insulin secretion that are not reflected by a change in glucose tolerance due to compensation outside the β-cell that occurs over time in vivo.

– "Since amino acids do not generate FBP, which is needed to allosterically activate PKm2, PKm1 is necessary to raise ATP/ADPpm, close KATP channels, and stimulate insulin secretion in response to amino acids." I don't recall that the authors measured plasma membrane (PM)-localised ATP/ADP rises. A PM targeting sequence might be something to add to Perceval-HR in the future.

This is an excellent suggestion and an avenue that we plan to pursue in the future.

– Male mice are used throughout the study. Either the authors should confirm major findings in female mice, or discuss this limitation in the text.

This has been noted in the last paragraph of the discussion.

– Please show full unedited Western blots in the Supplementary Information, as well as describe the loading controls e.g. was this the same lanes, parallel lanes or a separate blot?

Per the standard for *eLife* submissions, we provided the unedited Western blots in our source data files with the original submission. The reviewer should have access to these files, which show that the loading controls were performed on the same blot.

Reviewer #3 (Recommendations for the authors):In Figures 1B and 1D the western images do not represent the quantification bar graphs. PKm1 seems completely absent in 1B and same for Pkm2 in 1D.

Many thanks to the reviewer for pointing out this error. We mistakenly reported the Western blot intensities without background correction for the PKm1-βKO and PCK2-βKO, which underestimated the efficiency of knockdown in these two models. We have updated the figures and text accordingly.